# Photocatalytic Reduction of Carbon Dioxide on TiO_2_ Heterojunction Photocatalysts—A Review

**DOI:** 10.3390/ma15030967

**Published:** 2022-01-26

**Authors:** Beatriz Trindade Barrocas, Nela Ambrožová, Kamila Kočí

**Affiliations:** Institute of Environmental Technology, CEET, VSB-Technical University of Ostrava, 17 listopadu 15/2172, 70800 Ostrava, Czech Republic; beatriz.trindade.martins.vidigal.barrocas1@vsb.cz (B.T.B.); nela.ambrozova@vsb.cz (N.A.)

**Keywords:** CO_2_ reduction, heterojunction nanocomposite, TiO_2_, photocatalysis, renewable fuel, valuable chemicals

## Abstract

The photocatalytic reduction of carbon dioxide to renewable fuel or other valuable chemicals using solar energy is attracting the interest of researchers because of its great potential to offer a clean fuel alternative and solve global warming problems. Unfortunately, the efficiency of CO_2_ photocatalytic reduction remains not very high due to the fast recombination of photogenerated electron–hole and small light utilization. Consequently, tremendous efforts have been made to solve these problems, and one possible solution is the use of heterojunction photocatalysts. This review begins with the fundamental aspects of CO_2_ photocatalytic reduction and the fundamental principles of various heterojunction photocatalysts. In the following part, we discuss using TiO_2_ heterojunction photocatalysts with other semiconductors, such as C_3_N_4_, CeO_2_, CuO, CdS, MoS_2_, GaP, CaTiO_3_ and FeTiO_3_. Finally, a concise summary and presentation of perspectives in the field of heterojunction photocatalysts are provided. The review covers references in the years 2011–2021.

## 1. Introduction

Since the 18th century, together with the fast development of human society and the extensive use of fossil energy, environmental pollution has become increasingly serious with great environmental, social and economic impacts. Emissions of CO_2_ and other greenhouse gases are steadily increasing. The anthropogenic source of greenhouse gas emissions is fossil fuels combustion, mainly coal, natural gas, and oil, along with soil erosion and deforestation. These gases warm the planet by trapping heat in the atmosphere and are the principal factor in global warming. The average increase in temperature since the preindustrial age has already reached almost one degree Celsius, and this rise looks set to continue [1].

The chance of CO_2_ transformation into clean fuel could provide a progressive solution for both the future deficiency of fossil fuels and problem with global warming. In recent years, a large number of methods, such as chemical, electrochemical, biochemical, photochemical, and thermochemical techniques, have been developed for converting CO_2_ to light hydrocarbons and alcohols [2,3]. Among the varied methods, CO_2_ photocatalytic reduction has been receiving great attention and proved to be a promising alternative technology, once it is possible to produce greener gases and/or gases with industrial and fuel applications, using sunlight to activate the semiconductor materials, which result in the photoreduction of gaseous pollutants. This method is one of the promising processes, which not only remove carbon dioxide, but can also transform it into energy valuable products, such as methane, formaldehyde, methanol, CO and other useful compounds [4,5,6,7].

Photocatalysis can be defined as a change in the rate of a photochemical reaction by the activation of a photocatalyst (semiconductor) with sunlight or artificial light (ultraviolet or visible radiation). This process is very efficient and attractive from the economical and eco-friendly point of view. This method is based on the use of a photocatalyst, usually a semiconductor, illuminated with energy equal to or higher than its energy of the bandgap.

It is well known that CO_2_ is an extremely stable molecule with high thermodynamic stability, being that its reduction is extremely complicated. The photocatalytic reduction of CO_2_ has complex reaction mechanisms and pathways involving a proton-assisted multi-electron reduction process with high energy barriers, complicated activation and CO_2_ adsorption, and selectivity of different products as shown in Table 1.

The photocatalytic system for the reduction of CO_2_ makes use of a photocatalyst suspension in a solvent with dissolved carbon dioxide, and irradiation with solar energy can drive the photoreduction of CO_2_. Hole scavengers, such as H_2_O_2_, Na_2_SO_3_/Na_2_S, CH_3_OH, and triethanolamine, are ordinarily added to the reaction mixture to decrease the electron–hole recombination and avoid reoxidation by generated holes or the oxygen which is produced from water [9].

Inoue et al. reported, in 1979, CO_2_ photocatalytic reduction using several semiconductors (dispersed in water) as photocatalysts. They studied TiO_2_, WO_3_, CdS, ZnO, GaP, and SiC for the photocatalytic reduction of CO_2_, and concluded that TiO_2_ and SiC materials had higher photocatalytic activity for this reaction [11]. Recently, several photocatalysts, such as TiO_2_, g C_3_N_4_, ZnIn_2_S_4_, Bi_2_WO_6_, graphene (GR), CdS, SrNb_2_O_6_, and ZnO, were investigated for CO_2_ photocatalytic reduction. However, the TiO_2_ is the most prevalently used due to its chemical stability, resistance toward corrosion, and mainly low cost [12].

TiO_2_ has naturally three polymorphic phases: brookite, anatase, and rutile [13]. TiO_2_ has a large band gap and, therefore, the solar light utilization rate of TiO_2_ is only 4%. Therefore, the photocatalytic performance of TiO_2_ using solar energy is limited [14]. TiO_2_ has a relatively high recombination rate of photoinduced electron/hole (e^−^/h^+^) pairs [10]; hence, only a fraction of the generated e^−^/h^+^ pairs are available for photoreaction [15,16,17]. In recent years, there has been an effort to increase the photocatalytic activity of TiO_2_. Several strategies have been suggested to efficiently separate pairs of photogenerated electrons and holes in semiconductor photocatalysts, thereby increasing the efficiency of photocatalysis. Some of the most important ones are, for example, doping metals or non-metals, or creating photocatalysts with heterojunction. The formation of heterojunction photocatalysts, where the generated electron–hole pairs are efficiently separated, has emerged as one of the most promising approaches (Figure 1).

A heterojunction is the interface between two diverse materials which has a different band structure, and it can lead to band alignments. Many types of heterojunctions have been studied that are efficient for increasing the photoactivity of materials. These include conventional heterojunctions (type-I, type-II and type-III), surface heterojunctions, p–n heterojunctions, direct Z-scheme heterojunctions, and graphene-semiconductor (graphene-SC) heterojunctions [18]. In the conventional heterojunction photocatalysts, there are three types: the type-I have a straddling gap (Figure 2a), type-II have a staggered gap (Figure 2b), and the type-III a broken gap (Figure 2c).

Figure 2a shows the type-I heterojunction photocatalyst. The valence band (VB) and the conduction band (CB) of the semiconductor A are, respectively, lower and higher than the matching bands of semiconductor B. For that reason, after irradiation, e^−^ and h^+^ cumulate at the CB and the VB levels of the semiconductor B, which has lower Eg. Since both e^−^ and h^+^ cumulate on one and the same semiconductor, they cannot be effectively separated; therefore, it is not suitable for application in photocatalysis. Figure 2b represents the type-II heterojunction photocatalyst. The VB and the CB levels of semiconductor A are higher than the matching VB and CB levels of semiconductor B. In this case, the migration of photogenerated charges can occur in opposite directions, namely, the e^−^ are accumulated in one semiconductor, while the h^+^ are accumulated in the other semiconductor, resulting in a spatial separation of e^−^/h^+^ pairs. This separation prevents the rapid recombination of photogenerated charges. A semiconductor with appropriate band positions acts as a scavenger of e^−^ and h^+^, allowing these charges to react separately. The type-III, as can be seen in Figure 2c, has an architecture similar to the type-II heterojunction photocatalyst; however, there is no overlapping of band gaps, thereby being inadequate for photocatalytic applications [18,19].

The p–n heterojunction (Figure 3) combines the p-type semiconductor and n-type semiconductor. The Fermi level is closer to the valence band in a p-type semiconductor. On the other hand, in the case of an n-type semiconductor, it will shift toward the conduction band [18]. This configuration can increase migration of the electron–hole through the heterojunction for increasing the photocatalytic efficiency by giving an additional electric field. In this type of heterostructure, before light irradiation, the e^−^ on the n-type semiconductor diffuse across the p–n interface to the p-type semiconductor, abandoning positive holes (h^+^). In the meantime, the positive holes of the p-type semiconductor diffuse into the n-type semiconductor, abandoning negative electrons. This diffusion of electrons and holes continues until the Fermi levels of the semiconductors are equal. As a result, an internal electric field is formed on the p–n interface. The electrons and holes, which are photogenerated in p-type and n-type semiconductors, travel due to the impact of the internal electric field from the conduction band of the n-type to the valence band of the p-type, respectively, following the spatial separation of electrons and holes, and prolong their lifetime. Consequently, the efficiency of e^−^/h^+^ separation in the case of the p–n heterojunction is quicker than that of type-II heterojunction photocatalysts because of the synergic effect of the band alignment and the internal electric field [18]. For instance, it is very often that the combination of TiO_2_ (n-type) with a p-type semiconductor for the formation of a p–n heterojunction occurs [10].

However, for these types of heterojunctions mentioned above (conventional type-II and p–n heterojunction types), the redox capability of the photocatalyst is decreased, due to the oxidation and reduction processes take place on the semiconductor with lower oxidation and reduction potentials, respectively [10,18].

Another type of heterojunction is the Z-scheme photocatalytic system. The Z-scheme system for a liquid phase was reported in 1979 by A. J. Bard [20]. Since this discovery, the Z-scheme heterojunctions have become one of the major topics of interest for scientific researchers, to overcome the problems of the abovementioned heterojunction photocatalysts, such as the redox ability of the material [18].

The conventional Z-scheme photocatalytic system is formed with two semiconductors (PS I and PS II), which are not in physical contact, and a dissolved redox mediator consisting of an electron acceptor/donor (A/D) pair (Figure 4a). During the photocatalytic reaction, photogenerated electrons migrate from the CB of the PS II to the VB of the PS I through an A/D pair via following redox reactions.

The conventional Z-scheme photocatalytic system is formed with two semiconductors (PS I and PS II). Unfortunately, this type of heterojunction photocatalyst has the one limitation; they can solely be used in the liquid phase, in which they are not in physical contact, and an electron acceptor/donor (A/D) pair (Figure 4a), named the redox mediator. In this case, both oxidative and reductive photocatalysts are photoexcited, producing electrons and holes. After that, an e^−^ photogenerated in the oxidative photocatalyst reacts with the A, forming an electron donor (D) (Equation (1)). Furthermore, a hole in the reductive photocatalyst reacts with the D, producing an electron acceptor (Equation (2)).
(1)$$Acceptor+e^{-}\to Donor$$
(2)$$Donor+h^{+}\to Acceptor$$

Therefore, the acceptor (A) is reduced to a donor (D) when it reacts with the electrons from the conduction band of the photocatalyst I. Then, the donor (D) is oxidized and produces the acceptor (A) due to the reaction with the holes from the valence band.

In this type of heterojunction, electron–hole separation and a redox ability is achieved, due to the fact that electrons are cumulated in photocatalyst I, with higher reduction potential, while holes are cumulated in photocatalyst II, with higher oxidation potential. The conventional Z-scheme photocatalysts can only be constructed in the liquid phase, thereby limiting their wide application in photocatalysis [18,19].

Later, in 2006, Tada et al. [21] suggested a solid-state Z-scheme photocatalytic system consisting of two photocatalysts (PS I and PS II) connected by a solid-phase electron mediator. This mediator can lead the electrons to go from the oxidative photocatalyst to the reductive photocatalyst, eliminating the inactive charge carriers [19]. Furthermore, this system (Figure 4b) can be applied in practically all experimental conditions, markedly extending it using. However, noble metals (such as Pt, Ag, and Au), which are rare and expensive, are used usually as mediators of electrons in this system, being a limitation to their practical application. In addition, this type of mediator can also absorb incident light, decreasing the photocatalyst’s light utilization [18,19].

In 2013, Yu et al. [22] suggested a heterojunction photocatalyst with the direct Z-scheme. In this case, there is a combination of two different photocatalysts, without an electron mediator. Figure 4c shows that the construction of this direct Z-scheme is the same as the all-solid-state Z-scheme, except that the rare and expensive electron mediators are not required in this system [18,19]. Similarly, e^−^ and h^+^ are spatially separated on the material with the higher reduction potential and oxidation potential of the direct Z-scheme heterojunction photocatalyst, respectively. The fabrication cost of this direct Z-scheme is low and comparable to that of conventional type-II heterojunction systems. Furthermore, the electron–hole transfer on the direct Z-scheme heterojunction is physically more favorable than that on the type-II heterojunction due to the electrostatic attraction between electrons and holes. In particular, in the case of the direct Z-scheme photocatalysts, the transfer of e^−^ from the CB of the PS II to the h^+^ rich VB of the PS I is easier, due to the electrostatic attraction between the electrons and the holes. Moreover, without the use of liquid-phase or noble metal electron mediators, the direct Z-scheme photocatalysts have greater potential for wide practical applications [18,19].

The structure of a direct Z-scheme catalyst and p–n heterojunction is similar to that of a type-II heterojunction catalyst. For that reason, it is essential to study the charge-carrier migration mechanism for the different types of heterojunction photocatalysts through various characterization methods, so as to differentiate them. Therefore, various characterization methods could be used for this purpose, such as radical scavenging, photocatalytic reduction testing, metal loading, X-ray photoelectron spectroscopy (XPS), effective mass calculation, internal electric-field simulation and effective mass calculation. Using only a single characterization method cannot provide exact information on the charge-carrier migration mechanism for the heterojunction photocatalyst. Therefore, a comprehensive investigation through a combination of different characterization methods is always essential to describe the type of heterojunction photocatalysts [10,18].

In this review, the most promising semiconductors with heterojunction with TiO_2_ photocatalysts for CO_2_ photoreduction, such as C_3_N_4_ [23,24,25,26,27,28,29,30,31,32,33], CeO_2_ [34,35,36,37,38], CuO [39,40,41,42,43], CdS [44,45,46,47], MoS_2_ [48,49,50,51], and others [52,53,54], are summarized.

## 2. TiO_2_ Heterojunction Photocatalysts

### 2.1. g-C_3_N_4_/TiO_2_

Graphitic carbon nitride (g-C_3_N_4_) is a metal-free organic semiconductor, with special physicochemical properties, such as photocatalytic stability [55], electronic band structure, sufficient negative potential of conduction band, chemical and high thermal stability, and low cost. Due to its optical bandgap size (~2.7 eV), it can be activated by visible light, being an appropriate solar light harvesting photocatalyst [56,57,58]. However, this has some disadvantages that reduce its photocatalytic activity, such as high recombination of photogenerated charge carriers, low surface area, and low electrical conductivity [59]. These disadvantages can be overcome by combining them with other heterojunction semiconductors. The combination of wide-band TiO_2_ and small-band g-C_3_N_4_ as a visible light sensitizer to create a heterojunction structure can mask the light response of both photocatalysts, due to the special electronic band structure [60,61]. For this reason, we can harvest more light of the sun through a coupling of g-C_3_N_4_ and TiO_2_, forming a g-C_3_N_4_/TiO_2_ heterojunction. In addition to the CO_2_ photocatalytic reduction, the resulting heterojunction between TiO_2_ and C_3_N_4_ is used, for example, in the photocatalytic oxidation of NO [62], and for organic pollutants degradation in waste water [63]. CO_2_ photocatalytic reduction using g-C_3_N_4_/TiO_2_ heterojunction photocatalysts are tabulated in Table 2.

Shi et al. [25] reported yTiO_2−x_/g-C_3_N_4_ heterojunction photocatalysts with efficient solar-driven CO_2_ reduction. The 0D/2D heterostructure of oxygen vacancy-abundant TiO_2_ quantum dots referred in the g-C_3_N_4_ (MCN) nanosheets (TiO_2−x_/g-C_3_N_4_), were synthesized by in situ pyrolysis of NH2-MIL-125 (Ti) and melamine with their different mass ratios (g/g) of 5:0.4, 5:0.15, 5:0.1, and 5:0.05. The samples were named yTiO_2−x_/MCN (y = 8, 3, 2 and 1, which is identical to the % of Ti-MOF out of melamine). All yTiO_2−x_ /MCN photocatalysts showed magnificent photocatalytic reduction performance of CO_2_ compared with MCN. The authors concluded that the overall rapid decay of electron–hole pairs was ascribed to the interfacial charge transfer, which was attended by relaxation of recombination mediated by shallow trapped sites. Extremely fast interfacial charge transfers significantly increased charge separation. Thus, e^−^ in shallow trapped sites could be readily trapped by carbon dioxide. Moreover, coupling with the synergetic advantage of powerful visible light absorption, high adsorption of CO_2_ and large specific surface area, TiO_2−x_/g-C_3_N_4_ demonstrated an excellent CO evolution rate. This research shows detailed insights into optimizing the heterojunction structure for robust solar CO_2_ conversion. The 2TiO_2_-MCN performed the highest CO formation, roughly five times that of parent g-C_3_N_4_. These results show that the significant photoreduction performance of CO_2_ is also connected with the unique structures, and interface composition of the 0D/2D structure, such as defects in the photocatalyst as well as high specific surface area for enhancing CO_2_ adsorption and supporting charge carrier separation. In these experiments, only carbon monoxide and a small amount of hydrogen were detected [25].

Reli et al. [26] studied the TiO_2_/g-C_3_N_4_ materials with the ratio of TiO_2_ to g-C_3_N_4_ ranging from 0.3/1 to 2/1 (TiO_2_/g-C_3_N_4_ ratio of 0.3/1, 0.5/1, 1/1, and 2/1) for the photoreduction of CO_2_ and photoreduction of N_2_O. They reported that the production rate of methane is almost linear during the first 8 h of irradiation; on the other hand, the carbon monoxide yields increased rapidly in the first two hours and then are almost constant. The hydrogen was also detected. The hydrogen is generated from the water splitting. The most photoactive photocatalyst was (0.3/1)TiO_2_/g-C_3_N_4_, in the presence of which they observed the highest yields of the products. On the other hand, the lowest product formation was achieved over pristine g-C_3_N_4_. The authors concluded that the highest photoactivity of the (0.3/1)TiO_2_/g-C_3_N_4_ photocatalyst can be clarified by the combination of several aspects, such as adsorption edge energy, surface area (S_BET_), crystallite size and efficient charge carrier separation. The key parameter is the efficient charge separation [26]. 

Zhang et al. [27] described the synthesis of hollow TiO_2_@g-C_3_N_4_ nanocomposites for CO_2_ photocatalytic reduction under visible irradiation. In this work was reported the utilization of four TiO_2_@g-C_3_N_4_ composites with different mass ratios of g-C_3_N_4_ with respect to composites of 11.1%, 14.3%, 20% and 33.3%, labeled as TiO_2_@g-C_3_N_4_-11.1%, TiO_2_@g-C_3_N_4_-14.3%, TiO_2_@g-C_3_N_4_-20%, and TiO_2_@g-C_3_N_4_-33.3%, respectively. The results indicated that TiO_2_@g-C_3_N_4_ photocatalysts displayed higher photocatalytic activity, compared with pure g-C_3_N_4_ and the TiO_2_ does not showed photocatalytic activity under visible light irradiation. The increased photocatalytic activity of TiO_2_@g-C_3_N_4_ nanocomposites was attributed to the higher photo-induced electron–hole separation efficiency and enhanced photoinduced electron migration. Furthermore, the sample with the best photocatalytic performance for the CO_2_ reduction was the TiO_2_@g-C_3_N_4_-20%. They concluded that with the decrease in g-C_3_N_4_ content, the yield of methanol decreased, due to the fact that TiO_2_ has no catalytic activity in visible light, so the higher amount of TiO_2_ weakened the absorbing ability of visible light and reduced the photocatalytic efficiency of the TiO_2_@g-C_3_N_4_-11.1% and TiO_2_@g-C_3_N_4_-14.3% composite materials [27].

Dehkordi et al. [28] reported a hierarchical g-C_3_N_4_@TiO_2_ hollow sphere with brilliant activity for CO_2_ photocatalytic reduction under visible irradiation. These samples have TiO_2_ shell onto the surface of hollow carbon nitride sphere (HCNS) and are named HCNS@TiO_2_. In this work, the photocatalytic efficiency of the HCNS@TiO_2_ photocatalyst was compared with the commercial photocatalyst TiO_2_ (P25), g-C_3_N_4_ and P25/g-C_3_N_4_. The obtained results showed that the P25/g-C_3_N_4_ and HCNS@TiO_2_ samples had a superior efficiency for the conversion of CO_2_ to valuable products under visible light irradiation, once P25 and pristine g-C_3_N_4_ showed the small yield of CH_3_OH production due to the poor visible light activity and the fast rate of electron–hole recombination, respectively. Furthermore, the heterojunction photocatalyst formed through the combination of TiO_2_ and g-C_3_N_4_ with a special hierarchical hollow structure (HCNS@TiO_2_) showed to be promising with higher potential than each pristine photocatalysts in the CO_2_. In this reaction, methanol was the primary product in the beginning of irradiation; consequently, the authors concluded that the solar fuel (CH_3_OH/CH_4_) can be obtained by the control of the reaction time of CO_2_ photoreduction. The authors concluded that the nanocomposite photocatalytic activity could be ascribed to its special structure, providing properties, such as multiple light reflection, light harvesting, and an improved active site. They also observed that the improvement in the photocatalytic performance of the HCNS@TiO_2_ was obtained due to the increased light absorption. The efficiency of CO_2_ photoreduction over the HCNS@TiO_2_ photocatalyst was approximately 5 and 10 times higher than in the presence of pristine g-C_3_N_4_ and P25, respectively [28]. The decisive parameter responsible for the increasing the photocatalytic performance of HCNS@TiO_2_ photocatalyst is the synergistic heterojunction creation between the hollow g-C_3_N_4_ sphere with TiO_2_, which makes a rapid electron transfer at the interface between HCNS and TiO_2_ and increases charge carrier separation [28].

Furthermore, the interest in studying the efficiency of heterojunction materials for CO_2_ photoreduction has been increasing, and some studies have also been reported using the combination of g-C_3_N_4_ with doped TiO_2_, for instance, TiO_2_ doped with amine species (N), and modified with metals, for example Ag and Au. Therefore, the most relevant reported works are mentioned here in this review.

For instance, Zhou et al. described the selective photoreduction of carbon dioxide to CO, using the graphitic carbon nitride (g-C_3_N_4_)-N-TiO_2_ heterostructure as an effective photocatalyst [29]. In this work, the authors prepared photocatalysts of graphitic carbon nitride and in situ N-modified titanium dioxide (g-C_3_N_4_-N-TiO_2_ composites), using precursors that incorporate urea and Ti(OH)_4_ with various mass ratios (80:20, 70:30, 60:40, 50:50, 40:60, 30:70, 20:80, and 10:90). The greater ratios of urea to Ti(OH)_4_ (60:40 and more) result in the photocatalysts of g-C_3_N_4_ and N-doped TiO_2_, while smaller ratios (till 50:50) only show in N-doped TiO_2_. The selectivity of the photocatalytic reaction is interesting in the presence of these photocatalysts. In the presence of N-doped TiO_2_ samples, CH_4_ and CO were generated, while in the presence of g-C_3_N_4_ and N-TiO_2_, only CO was performed; the product selectivity may connect with the formed g-C_3_N_4_. Among the as-prepared samples, 70:30 g-C_3_N_4_ and N-TiO_2_ composites present the highest CO formation, due to the visible light absorption and lowest electrons and holes recombination. As can be seen in the CO_2_ photoreduction reactions in Table 1, eight electrons are required for the formation of one CH_4_ molecule; however, only two e^−^ are necessary for one CO molecule production. For that reason, the CO_2_ photoreduction into CO is a more dynamic, favored process.

Based on this fact and with the obtained results, the authors designed a mechanism for the increase in photocatalytic performance, where charge carriers are created and transmitted between the interface of g-C_3_N_4_ and N-TiO_2_ during irradiation. Therefore, the holes in g-C_3_N_4_ (h^+^ created in g-C_3_N_4_ and transmitted from the valence band of TiO_2_) might oxidize the H_2_O absorbed on the surface of g-C_3_N_4_, producing O_2_ and H^+^. Furthermore, the electrons in N-TiO_2_ (created in N-TiO_2_ and the e^−^ transmitted from g-C_3_N_4_) can reduce the CO_2_ into C1 intermediates. No methane was produced when a high ratio of urea and Ti(OH)_4_ (60:40 or more) was used. This was due to the presence of g-C_3_N_4_ and N-TiO_2_; the H^+^ may not capture the photogenerated e^−^, due to the formation of aromatic heterocycles of g-C_3_N_4_, which are electron rich, where the protons can be stabilized by the conjugated aromatic heterocycles and, thus, they have difficulty in taking part in the formation of CH_4_. Furthermore, the e^−^ in the conduction band of g-C_3_N_4_ can quickly be transferred to the conduction band of N-TiO_2_ for CO_2_ photoreduction into CO. For that reason, the g-C_3_N_4_-N-TiO_2_ photocatalyst formed is selective for the production of CO during the CO_2_ photoreduction. On the other hand, for low ratios of urea to Ti(OH)_4_ (till 50:50), the H^●^ radicals or H^+^ ions formatted during CO_2_ photocatalytic reduction can be quickly consumed by adsorbed carbon dioxide; thus, CO and CH_4_ were simultaneously analyzed, due to the absence of conjugated aromatic system on these samples [29].

Another example of the combination of g-C_3_N_4_ with doped TiO_2_ was reported by Truc et al. [30], using TiO_2_ doped with niobium. Truc et al. [30] studied the photoactivity of niobium doped TiO_2_/g-C_3_N_4_ direct Z-scheme photocatalytic system for effective CO_2_ conversion into valuable fuels. They prepared three Nb-TiO_2_/g-C_3_N_4_ samples with 25%, 50%, and 75% of the mole percentages of Nb-TiO_2_.

The authors observed that g-C_3_N_4_ did not reduce CO_2_ under visible light irradiation, due to the high recombination rates of photoexcited e^−^ and h^+^. However, CO_2_ photocatalytic reduction under visible irradiation was possible in the presence of Nb-TiO_2_ and Nb-TiO_2_/g-C_3_N_4_ materials, obtaining different products (CH_4_, CO, and HCOOH). In the presence of a pure Nb-TiO_2_ photocatalyst, the products were CO and CH_4_. Nb dopant in TiO_2_ lattice led to the creation of the Ti^3+^, which was as an intermediate band between the valence band and the conduction band of the TiO_2_, reducing the e^−^ and h^+^ recombination. Furthermore, when the Nb-TiO_2_/g-C_3_N_4_ was used as photocatalyst, not only CO and CH_4_ were produced, but also HCOOH was obtained [30].

As expected, the photocatalytic activity for CO_2_ reduction was higher for the Nb-TiO_2_/g-C_3_N_4_ samples, when compared to the Nb-TiO_2_, g-C_3_N_4_ and TiO_2_ samples. The authors attributed this to the direct Z-scheme mechanism, where photoexcited e^−^ in the Nb-TiO_2_ CB combined with the photoexcited h^+^ in the g-C_3_N_4_ VB avoided the existence of e^−^ in the g-C_3_N_4_ CB and h^+^ in the Nb-TiO_2_ VB. Therefore, this Nb-TiO_2_/g-C_3_N_4_ system had more available electron–hole pairs when compared with the pure Nb-TiO_2_ photocatalyst. Furthermore, the potential energy of the generated electrons of Nb-TiO_2_/g-C_3_N_4_, (~−1.2 V) was more negative than the generated electron of Nb-TiO_2_, (~−0.2 V), so the generated electron of the Nb-TiO_2_/g-C_3_N_4_ required lower energy during the reduction of CO_2_ when compared with that of the Nb-TiO_2_ [30].

The best photocatalyst for the photoreduction in CO_2_ under visible light irradiation was the 50Nb-TiO_2_/50 g-C_3_N_4_. The higher efficiency of this sample was due to the higher numbers of produced and consumed e^−^ and h^+^ when compared with the Nb-TiO_2_ and the other Nb-TiO_2_/g-C_3_N_4_ photocatalysts. In the 50Nb-TiO_2_/50 g-C_3_N_4_ sample, the Nb-TiO_2_ mole resembled the mole of g-C_3_N_4_; therefore, the photogenerated electrons in the Nb-TiO_2_ CB would achieve photogenerated holes in the g-C_3_N_4_ VB. Therefore, the amounts of e^−^ in the g-C_3_N_4_ CB and h^+^ in the Nb-TiO_2_ VB of the 50Nb-TiO_2_/50 g-C_3_N_4_ sample were considerably higher than in the presence of other photocatalysts. Based on this work, they concluded that the Nb-TiO_2_/g-C_3_N_4_ photocatalysts have more charge carriers available for different valuable fuels. Additionally, the produced electrons of the Nb-TiO_2_/g-C_3_N_4_ in the conduction band of the g-C_3_N_4_, for which the potential energy is around—1.2 V, are enough strong to produce not only CO and CH_4_, but also HCOOH during the reduction of CO_2_ [30].

Li et al., for example, used the heterostructured g-C_3_N_4_/Ag-TiO_2_ photocatalyst for the CO_2_ photocatalytic conversion [31]. These authors reported for the first time the preparation of heterostructured g-C_3_N_4_/Ag-TiO_2_ materials via a facile solvent evaporation and by a calcination process with g-C_3_N_4_ and Ag-TiO_2_ as precursors. They prepared four different g-C_3_N_4_/Ag-TiO_2_ samples with various masses of g-C_3_N_4_ and Ag-TiO_2_ and compared them with the commercial TiO_2_ photocatalyst (Degussa P25), g-C_3_N_4_, and AgTi samples. As expected, the results showed that TiO_2_ obtained the lower CO_2_ conversion, and no significant amount of CH_4_ was formed during the 3 h irradiation. Using g-C_3_N_4_, the CO_2_ conversion was higher in comparison with TiO_2_; however, the CH_4_ yields were still very low. The AgTi sample showed higher performance than TiO_2_, due to the Ag nanoparticles (NPs) on the AgTi sample not only making the separation of generated charge carriers on TiO_2_ by UV irradiation easy, but also increasing the energy of trapped e^−^ through the Ag surface plasmon resonance effect with the visible light irradiation. Due to this fact, there were more e^−^ with higher energy for CH_4_ formation during CO_2_ reduction. Using the CN/AgTi composite samples, both the conversion of CO_2_ and solar fuel (CH_4_ and CO) yields enhanced with the higher amount of g-C_3_N_4_ to AgTi mass ratio from 0 to 8%. They also observed that the rate of electrons consumed was higher when the composite samples were used.

The results showed that the 8CN/AgTi sample (with g-C_3_N_4_ to AgTi mass ratio of 8%) obtained the highest photoconversion of CO_2_ after 3 h of irradiation. Based on this result, the authors concluded that the coupling of g-C_3_N_4_ and AgTi had a synergistic effect in the photocatalytic reduction of CO_2_. However, when the g-C_3_N_4_ to AgTi mass ratio increased to 12%, this led to in an evident decrease in the photocatalytic reduction of CO_2_; this decreasing trend is ordinary, and it is possible to attribute it to the fact that an excessive amount of g-C_3_N_4_ resulted in shielding of the active site on the TiO_2_ surface.

The authors reported that the combination of g-C_3_N_4_ and AgTi enhanced the generation of electrons and holes under sunlight. These photogenerated electrons moved through the heterojunction between carbon nitride and titanium dioxide, and further from titanium dioxide to silver nanoparticles with a lower Fermi level, avoiding the electron–hole recombination, and led to electrons cumulating on Ag nanoparticles deposited on the surface of TiO_2_ in the g-C_3_N_4_ /Ag TiO_2_. After that, the e^−^ cumulated on the Ag nanoparticles were further energized by the surface plasmon resonance effect. Therefore, the CN/AgTi samples showed higher photocatalytic performance. 

The Ag nanoparticles on the TiO_2_ surface in the CN/AgTi composite had a significant role once they decelerated the e^−^/h^+^ pairs recombination due to extracting e^−^ from conductive band of TiO_2_, and also used the surface plasmon resonance effect to increase the energy level of e^−^ cumulating on the surface. Therefore, the bounteous energetic electrons on Ag nanoparticles generated from the activation by solar irradiation of the both TiO_2_ and g-C_3_N_4_ parts were answerable for the important synergy of the combination of g-C_3_N_4_ and AgTi in photoreduction of CO_2_ in the presence of water vapor [31].

Liu et al. studied the P–O functional bridges effects on the electron and hole transfer and separation of TiO_2_/g-C_3_N_4_ photocatalysts for the CO_2_ reduction [32]. The authors prepared four P–O bridge TiO_2_/g-C_3_N_4_ composite samples with various molar % ratios of phosphate to TiO_2_ (1, 5, 10 and 15%) and compared them with the g-C_3_N_4_ and TiO_2_/g-C_3_N_4_ samples. All photocatalysts exhibited excellent photocatalytic activity for CO_2_ reduction, being that CH_4_ was the principal product obtained as well as CO in a small amount. The sample TiO_2_/g-C_3_N_4_ with the P–O bridge with 10% molar ratio showed the best performance for this reaction, with photoactivity approximately 2 and 3 times higher than for pure samples. The P–O functional bridges increased the heterojunction coupling between TiO_2_ and g-C_3_N_4_, thereby significantly enhancing the charge transfer and separation, obtaining higher photocatalytic activity.

Based on this study, the authors concluded that the photoactivity of g-C_3_N_4_ was significantly improved due to the connection with P–O-bridged TiO_2_ (in a proper amount). The characterization of the P–O bridge TiO_2_/g-C_3_N_4_ composite samples, using surface photovoltage and photoluminescence spectroscopy, showed that the improvement on the e^−^/h^+^ separation of g-C_3_N_4_ after coupling with the P–O bridged TiO_2_, resulted from the P–O bridge between TiO_2_ and g-C_3_N_4_ that promotes effectively the electrons’ transference from g-C_3_N_4_ to TiO_2_ [32].

Sun et al. [33] reported the preparation of a Z-scheme heterostructure with r-TiO_2_ (rutile) modified with gold and g-C_3_N_4_ quantum dots to achieve a recyclable and high-efficiency photocatalyst for CO_2_ reduction. The photocatalytic activity of (Au, C_3_N_4_)/TiO_2_ composite was compared with the C_3_N_4_/TiO_2_, r-TiO_2_, and bulk C_3_N_4_. The obtained products were CO, CH_4_ and O_2_; however, using the pristine r-TiO_2_ and bulk g-C_3_N_4_ samples, no significant CH_4_ yield was observed. The higher-energy products require a higher reduction potential. Carbon dioxide reduction to methane is an 8-electrons process, which requires the photogenerated charge to have a long lifetime. The CB band of the pure r-TiO_2_ is not negative enough to transfer CO_2_ to CH_4_. However, for the (Au, C_3_N_4_)/TiO_2_ composite, the yield amount of carbon monoxide and methane was markedly enhanced. The authors concluded that a Z-scheme heterostructure was formed at the r-TiO_2_ and g-C_3_N_4_ interface, instead of type-II heterojunction [33].

The results demonstrated that the (Au, C_3_N_4_)/TiO_2_ photocatalyst has four and five times higher photoactivity in comparison with the bulk g-C_3_N_4_ and pristine r-TiO_2_, respectively. This improvement on the photocatalytic performance was attributed to the excellent Z-scheme heterojunction formed in the interface of r-TiO_2_ and g-C_3_N_4_ [33].

### 2.2. CeO_2_/TiO_2_

Ceria or cerium oxide (CeO_2_) is a rare earth metal oxide that has attracted the interest of researchers due to the fact that the valences of ceria, such as Ce^4+^ and Ce^3+^, enhance light absorption ability and increase electron transfer. This material is an n-type semiconductor with a large bandgap energy (2.7–3.2 eV), non-toxic, readily available, and chemically stable [64]. CeO_2_ present two oxidation states, Ce (IV) and Ce (III), which give it unique chemical, mechanical, and magnetic properties. Furthermore, the Ce (III) and Ce (IV) oxidation states can be easily converted from one to another [64,65]. The surface oxygen vacancies of CeO_2_ from the reversible characteristics of Ce^3+^ and Ce^4+^ can promote its photocatalytic performance. During the reduction of Ce^4+^ ions into Ce^3+^, there occurs a formation of oxygen vacancies on the photocatalyst surface, which consequently act as electron trap centers that can inhibit the recombination [64]. Recently, the utilization of CeO_2_ as a coupling was reported since the Ce^4+^/Ce^3+^ displacement can accelerate the charge separation and impurity levels caused by CeO_2_ coupled TiO_2_ to be excited in the visible region [66]. CO_2_ photocatalytic reduction using CeO_2_/TiO_2_ heterojunction photocatalysts are tabulated in Table 3.

Wang et al. [34] prepared three photocatalysts with various Ce/Ti molar ratios, 1:2, 1:1, and 2:1. All prepared CeO_2_-TiO_2_ composites had higher photoactivity for the CO_2_ photoreduction to CH_4_ and CO, when compared with Mes-CeO_2_, Mes-TiO_2_ and commercial TiO_2_ photocatalyst (P25). This enhancement of the photocatalytic efficiency for these CeO_2_-TiO_2_ photocatalysts was achieved due to the ordered large specific surface area, mesoporous architecture, 2D open-pore system that facilitates the diffusion of the reactant into the bulk of photocatalyst and consequently provides fast intraparticle molecular transfer, and also due to the absorption in the visible range due to the CeO_2_ species photosensitization. The heterojunction between CeO_2_ and TiO_2_ also contributes to the enhancement of the CeO_2_-TiO_2_ composites, once the photogenerated e^−^ in the TiO_2_ can be transferred for the CeO_2_ under the internal electric field, improving the e^−^/h^+^ separation in the TiO_2_, leading to the improvement of photoactivity under irradiation [34].

The authors also confirmed by XPS analysis that the presence of CeO_2_ can increase significantly the chemisorbed of oxygen species on the surface of the ordered mesoporous CeO_2_-TiO_2_ composites. These O species can easily trap e^−^ and produce surface O• with outstanding reduction ability. Additionally, the mixture of Ce^3+^/Ce^4+^ oxidation states on the CeO_2_-TiO_2_ surface show that the partial metal in photocatalysts is not completely oxidized, and therefore, Ce^3+^ can react with holes and avoid the recombination of photogenerated e^−^/h^+^, leading to a higher quantum effectiveness of CO_2_ photoreduction [34].

Comparing the efficiency of the three different composites, no significant differences were obtained in the obtained yields of CO and CH_4_ after 325 min of irradiation. The authors analyzed the stability of the composites after irradiation and concluded that these composite were stable after the photocatalytic test [34].

Abdullah et al. [35] reported a CeO_2_-TiO_2_ composite for the photoreduction of CO_2_ into CH_3_OH under Vis irradiation. They demonstrated that the methanol yield in the presence of CeO_2_-TiO_2_ was three times higher than that of pure TiO_2_. The researchers concluded that this improvement is due to the existence of active anatase phase of titanium dioxide with a small crystalline size, and the uniform structure and smaller bandgap of the CeO_2_-TiO_2_ photocatalyst increased the visible light absorption and produced more e^−^/h^+^ pairs. Furthermore, the existence of both Ce^3+^ and Ce^4+^ oxidation states on the surface of CeO_2_-TiO_2_ avoided the recombination of the photogenerated e^−^/h^+^. In this case, the e^−^ are captured by the Ce^4+^, and these trapped electrons are moved to the adsorbed oxygen in order to produce superoxide anion radicals, while Ce^3+^ reacts with the generated h^+^, reducing the e^−^/h^+^ recombination. As can be seen in the schematic representation in Table 3, the CeO_2_ has a more negative CB energy, due to the possibility of photoexcited e^−^ transference from CB of CeO_2_ to CB of TiO_2_, decreasing the recombination rate of the charge carriers [35].

Jiao et al. [36] prepared four CeO_2_/TiO_2_-*n* photocatalysts, with the weight ratio of CeO_2_ to TiO_2_ of *n*/100, obtaining the samples CeO_2_/TiO_2_-16, CeO_2_/TiO_2_-8, CeO_2_/TiO_2_-4, and CeO_2_/TiO_2_-2, and compared the photoactivity of the samples with the 3D ordered macroporous TiO_2_ (3DOM) and the commercial TiO_2_ (P25). The outcomes demonstrated that the CeO_2_/TiO_2_-2, CeO_2_/TiO_2_-4 and CeO_2_/TiO_2_-8 composites had higher photoactivity than the TiO_2_ and P25 samples, showing that the synergetic effect of TiO_2_ and CeO_2_ increased the photocatalytic efficiency. The CeO_2_ sample showed the lower CO production amount. The best photocatalytic performance was obtained with the sample CeO_2_/TiO_2_-4. However, the amount of CO decreased with the increase in the CeO_2_ loading amount (>4), showing that the amounts of CeO_2_ nanolayers have optimal value. The composite sample with the higher amount of CeO_2_, CeO_2_/TiO_2_-16, showed lower performance than the 3DOM TiO_2_ and P25 samples, which can be explained due to the fact that the CeO_2_ sample almost did not have photocatalytic activity for this reaction condition, and this can be the possible reason for the lower photocatalytic activity of this sample. The authors proposed a type-II photocatalytic mechanism for the CO_2_ photoreduction using the CeO_2_/TiO_2_ composite, shown in Table 3. The increase in photoactivities during the photocatalytic reduction of CO_2_ under Vis irradiation is due to the synergistic effect of the heterojunction between CeO_2_ and TiO_2_ and photonic crystals. They concluded that the heterojunction between CeO_2_ and TiO_2_ increases the charge carriers separation, and the absorption efficiency of solar irradiation can be enhanced due to the slow light effect of the 3D ordered macroporous structure and the ordered macroporous facilitates the diffusion of the reactants [36].

Zhao et al. [37] investigated the effect of the TiO_2_ polymorph phases, brookite, anatase, and rutile on the CeO_2_/TiO_2_ composites efficiency for the photocatalytic reduction for CO_2_. They prepared CeO_2_/TiO_2_ composite using anatase, brookite and rutile, identified as A-TiCe, B-TiCe and R-TiCe, respectively. The results showed that the higher amount of CO yield produced was achieved using the sample rutile TiO_2_/CeO_2_ (R-TiCe). This enhancement in CO_2_ photoreduction using the rutile TiO_2_/CeO_2_ sample was justified due to the formation of Ti defects at the CeO_2_-rutile interfaces that improves the energy-band structure of rutile, facilitating the e^−^/h^+^ pairs’ separation. To go further, the authors prepared samples of rutile TiO_2_/CeO_2_ with different mass ratios of CeO_2_/TiO_2_, with CeO_2_ 5.9, 12.9 and 24.3 wt.%, obtaining the samples, R-TiCe_0.05_, R-TiCe_0.1_, R-TiCe_0.2_, respectively. They compared the activity of these rutile TiO_2_/CeO_2_ composites with the rutile TiO_2_, CeO_2_ and P25. The CeO_2_ had the lower photocatalytic activity. The best photocatalyst for the CO_2_ photoreduction was the R-TiCe_0.1_ with the yield of CO. This result can be explained once the activity of CeO_2_ is markedly lower when compared with the TiO_2_, suggesting that in the CeO_2_/TiO_2_ composites, the TiO_2_ is the principal active composition, and CeO_2_ acts as a promoter [37].

Wang et al. [38] synthesized CeO_2_/TiO_2_ samples with CeO_2_ 40, 20 and 10 wt.%, identified as 0.4 CeO_2_/TiO_2_, 0.2 CeO_2_/TiO_2_ and 0.1 CeO_2_/TiO_2_, respectively. The activity of the CeO_2_/TiO_2_ composites was compared with the CeO_2_ and TiO_2_ samples. Comparing all the samples, the best photocatalyst was the 0.2 CeO_2_/TiO_2_. Furthermore, they observed that the CeO_2_/TiO_2_ composites’ photoactivity enhances with the higher CeO_2_ amount and reaches a maximum at 20 wt.% CeO_2_ content (sample 0.2 CeO_2_/TiO_2_), since for the sample with higher CeO_2_ content, the photocatalytic activity decreased. TiO_2_ is the principal active composition, while CeO_2_ acts as a promoter in the CeO_2_/TiO_2_ composites. CeO_2_ content is the dominant factor on the enhancement of CO_2_ photoreduction under simulated sunlight illumination. This work showed that CeO_2_ extends the light absorption of the CeO_2_/TiO_2_ composite to the Vis range and enhances the e^−^/h^+^ separation, due to the presence of Ce^3+^ [38].

### 2.3. CuO/TiO_2_

Copper oxide, CuO, is an p-type semiconductor nanomaterial with a bandgap between 1.2 and 1.9 eV, among this narrow direct bandgap. This material has various properties, such as high electrical conductivity, good semiconducting nature, thermal stability, low toxicity and low cost. The bandgap of CuO should favor the Vis light absorption and enhance the photoactivity [41,67,68]. CuO has been used as photocatalyst for the CO_2_ photocatalytic conversion to solar fuels [69,70]. CuO exhibits spontaneous CO_2_ adsorption (ΔH= −45 kJ mol^−1^) in comparison with TiO_2_. The energy levels of the CO_2_-adsorbed species, such as –O–Cu–O–, can lead to an improvement in the visible-light absorption and efficient separation of electrons and holes that favors the photocatalytic activity of CuO [69,71]. Moreover, CuO presents selectivity to the formation of value-added solar fuels, such as CH_3_OH and CH_4_ in the photocatalytic CO_2_ reduction [72]. For the above reasons, CuO/TiO_2_ photocatalysts with heterojunction were also studied for the CO_2_ photocatalytic reduction. CO_2_ photocatalytic reduction using CuO/TiO_2_ heterojunction photocatalysts is tabulated in Table 4.

Zhao et al. successfully prepared CuO-incorporated TiO_2_ photocatalysts by an impregnation method, to be used as photocatalysts for the CO_2_ reduction into methyl formate in methanol [41]. They observed that the heterojunction photocatalyst CuO/TiO_2_(AB) had higher methyl formate yield than the pristine photocatalysts. It was caused by its mixed-phase heterojunction structure and higher specific surface area, resulting in an effective separation, an enhanced UV-light response, and a smaller recombination rate of photogenerated electrons and holes. CuO/TiO_2_(AB) also demonstrated sufficient stability. The methyl formate yield reproducibility was higher than 90% in cyclic runs [41]. The experimental conditions are resumed in Table 4.

Li et al. [42] dealt with CO_2_ photocatalytic reduction to produce CH_3_OH and C_2_H_5_OH over CuO-loaded titania powders suspended in H_2_O with Na_2_SO_3_, which was the hole scavenger and promoted the formation of ethanol. The authors prepared four composites with a copper amount between 1 and 7 wt.% (7 wt.% CuO/TiO_2_, 5 wt.% CuO/TiO_2_, 3 wt.% CuO/TiO_2_, and 1 wt.% CuO/TiO_2_). They observed that yields of methanol and ethanol are enhanced with a CuO amount until 3 wt.%, and for the samples with 5 and 7 wt.%, the yields are decreased, being in this case 3 wt.%, the ideal amount of CuO loading. Loading of CuO enhances CH_3_OH and C_2_H_5_OH yields due to the higher amount of active sites. Copper is an electron catcher and inhibits e^−^/h^+^ recombination. However, the samples with a higher amount of copper (>3 wt.% CuO) cannot further enhance the CH_3_OH and C_2_H_5_OH yields due to the excess of CuO, which covers the surface of TiO_2_, decreasing the TiO_2_ photoexciting capacity, thereby reducing the photoactivity [42].

Another example of CuO and TiO_2_ heterojunction was reported by Qin et al. [39]. They studied the photocatalytic reduction of carbon dioxide in CH_3_OH to methyl formate in the presence of CuO–TiO_2_ photocatalysts. The methanol was used as the hole scavenger, which can react with the photogenerated holes in the VB, and CO_2_ was reduced by the e^−^ in the VB. The authors prepared samples with 0.5, 1, 3 and 5 weight percentage of CuO and compared their photocatalytic activity with TiO_2_. The coupling of TiO_2_ with CuO led to the rapid increase in the photoactivity because TiO_2_ and CuO created composite photocatalysts, and electron and hole recombination was reduced.

However, as mentioned above, these authors also concluded that higher CuO loading (>1.0%, in this case) decreases the photoactivity because of the CuO particles’ agglomeration. The most active photocatalyst was 1.0CuO–TiO_2_ (1 wt.% of CuO). The authors concluded that the heterojunction between TiO_2_ and CuO was the decisive parameter for enhancing the photoactivity of the samples [39].

Razali et al. [43] prepared p–n type CuO-TiO_2_ nanotube samples with improved ability for carbon dioxide photoconversion into fuels. They concluded that the higher photocatalytic efficiency of CuO—TiO_2_ photocatalyst is attributed to the restraint of e^−^ photogeneration and h^+^ recombination, as the p–n heterojunction between the CuO particles and TiO_2_ nanotube facilitates the charge separation between electrons and holes, due to the presence of an electrostatic field at the junction. Electrons in the CB of CuO transfer into the CB of TiO_2_, whereas holes in the VB of TiO_2_ transfer to the VB of CuO. The charge transfers and separation between both semiconductors may prohibit the recombination of electrons and holes, thus increasing the photocatalytic performance of the CuO-loaded TiO_2_ nanotube [43].

### 2.4. CdS/TiO_2_

Cadmium sulfide (CdS) is a semiconductor material from the II–VI group with a direct bandgap of 2.4 eV [73,74]. CdS is used for carbon dioxide photocatalytic reduction under UV light irradiation [11,75]. This photocatalyst has ideal properties, such as the capability of converting light energy into chemical, optical, photophysical and photochemical energy [73]. However, this material has some disadvantages, such as fast e^−^/h^+^ recombination, and photocorrosion vulnerability in aqueous solution due to oxidation by photo-generated holes during photocatalytic reaction [73]. Nevertheless, the photocatalytic activity of this semiconductor can be enhanced, for instance, by doping with metal elements or by the combination with other semiconductors [73]. To date, CdS is widely used in the TiO_2_/CdS coupled heterojunction to improve the photoelectron conversion efficiency of photocatalysis and solar cell [74]. TiO_2_/CdS combination is reported as the one of the most representative hybrid semiconductors, once the valence and conduction bands of the CdS are appropriately located in relation to those of TiO_2_ for higher charge separation, and also CdS can absorb a main part of visible light, as it is possible to use sunlight [44,76]. CO_2_ photocatalytic reduction using CdS/TiO_2_ heterojunction photocatalysts is tabulated in Table 5.

Park et al. [44] reported the photocatalytic conversion of CO_2_ to CH_4_ in the presence of TiO_2_/CdS in an isopropanol (IPA) solution under UV-Vis and Vis light irradiation. IPA is frequently used as a sacrificial e^−^ donor, such as an h^+^ scavenger. The authors prepared three TiO_2_/CdS composites, TiO_2_/CdS-5, TiO_2_/CdS-3 and TiO_2_/CdS-1, with varied amounts of loaded CdS to TiO_2_, approximately 33.7%, 23.6%, and 11.4%, respectively. However, they did not observe a significant difference on the obtained yields with the CdS amount on the TiO_2_, so they only reported the results obtained using the sample TiO_2_/CdS-3 as a photocatalyst for the CO_2_ photoreduction. The authors analyzed the H_2_ evolution and the production of CO and CH_4_, using Ar or CO_2_ gas to purge (before irradiation) the aqueous TiO_2_ and TiO_2_/CdS suspensions with isopropanol.

During the H_2_ evolution, the results showed that using Ar-purged gas, the TiO_2_/CdS composite sample had better photocatalytic activity in comparison with the pristine TiO_2_. As expected, the production of H_2_ in the Ar atmosphere was higher than in the CO_2_ atmosphere, due to the competition for electrons. In contrast, for the CO production, it was observed that using CO_2_-purged gas, the TiO_2_/CdS composite sample had better photocatalytic activity in comparison with the pristine TiO_2_. This result was observed due to the better adsorption of CO_2_ on CdS, as well as the increased charge carrier separation and transfer on the TiO_2_/CdS composite sample. 

Regarding the CH_4_ production, the results showed that regardless of the purge gas used (Ar or CO_2_), the TiO_2_/CdS composite sample had higher catalytic activity when compared with the TiO_2_ sample. It is well known that CdS-modified TiO_2_ is more active than pure TiO_2_ for the formation of CH_4_. However, in this case, some part of the CH_4_ obtained was formed due to the presence of IPA, both making a contribution to the observed CH_4_ yields. Furthermore, the authors did not discard the idea that the presence of hydrocarbon contaminants during the preparation of the catalyst can be considered for a fraction of the obtained yields.

The authors also investigated the photoactivity of the TiO_2_/CdS sample (in CO_2_-purged gas) under Vis light irradiation. In this case, only the CdS photocatalyst was capable of excitation, and no significant differences on the CH_4_ production were obtained using UV-vis and visible light irradiation, suggesting that the CdS plays a significant role in CO_2_ fixation and in photocatalyzing the transference of multi-electrons to CO_2_.

With this work, the authors concluded that the presence of CdS on TiO_2_ enhanced the production rate of CH_4_ and enhanced the total CH_4_ yields. They reported that this enhancement can be ascribed to the easy transference of e^−^ from the CdS to surface-bound CO_2_, resulting in to the formation of ^•^CO_2_^−^ that binds to the positively charged surface of CdS, and also due to the surface-bound bicarbonate geometry that increases the production of CH_4_ due to smaller energy barriers in comparison with the linear O=C=O molecule [44].

Low et al. [45] described a direct Z-scheme TiO_2_/CdS composite with high efficiency for the photocatalytic reduction of CO_2_. They compared the photocatalytic activity of the TiO_2_/CdS composite with the TiO_2_, CdS and commercial P25 samples. The TiO_2_/CdS composite formed 3.5-, 5.4-, and 6.3-times higher amounts of CH_4_ than the TiO_2_, CdS and commercial P25, respectively. They compared the type II and direct Z-scheme possibility for the mechanism of their TiO_2_/CdS composites during activation. With a simple test of the ^●^OH production (using coumarin to trap ^●^OH and produce fluorescent products, it is possible to analyze by fluorescence spectroscopy) they concluded that it was possible to produce this radical. On the other hand, using the CdS photocatalyst was not obtained, due to the position of the VB (around 1.8 V) being lower than the potential of this reaction (*E*^0^(OH^−^/^●^OH) = 2.4 V). So, the ^●^OH was formed on the TiO_2_ side of the composite, following the direct Z-scheme mechanism of the charge-transfer process, as shown in the schematic illustration in Table 5. The enhanced performance obtained in the presence of the TiO_2_/CdS composite can be explained due to the e^−^/h^+^ availability (according to the enhanced photocurrent for this sample), due to the direct Z-scheme heterojunction [45].

Song et al. [46] prepared four CdS–TiO_2_ samples with various molar ratios, named CdS–TiO_2_-X (X is molar ratios of TiO_2_/CdS: CdS–TiO_2_-10, CdS–TiO_2_-9, CdS–TiO_2_-8, and CdS–TiO_2_-6. All composites had higher activity for the CO_2_ photoreduction than CdS and TiO_2_. This can be explained due to the interaction between TiO_2_ and CdS that improved the photocatalytic reduction capacity of CO_2_.

They observed that the increase in the CdS amount in the composites until 8:1 increased the efficiency, obtaining the CdS–TiO_2_-8, optimal photoactivity for the production of cyclohexyl formate (CF) and cyclohexanone (CH). However, for CdS content higher than 8:1 TiO_2_/CdS molar ratios, a decrease in the photocatalytic activity was observed, indicating that a high amount of CdS in the TiO_2_ photocatalyst decreases the photogenerated e^−^ on the TiO_2_ and then leads to a smaller photocatalytic activity. The exactly 1 mole of excited TiO_2_ has to correspond to 1 mole of excited CdS; otherwise, the exceeded e^−^ or h^+^ recombine to decrease the reaction rates. In addition, for the higher than 8:1 TiO_2_/CdS molar ratios, there were difficulties in the light absorbance once the higher amount of CdS aggregated on the surface of the TiO_2_ nanosheets, which hampered the absorption of light by the TiO_2_. Furthermore, they concluded that the CO_2_ absorbed in cyclohexanol can be decreased to CF, and the cyclohexanol oxidized into CH on the conduction band and valence band of the TiO_2_/CdS photocatalyst, respectively (as can be seen in the scheme of Table 5) [46].

Ahmad Beigi et al. [47] reported the preparation of CdS/TiO_2_ nanocomposites for the photocatalytic reduction of CO_2_ to CO and CH_4_ under UV-vis and visible light irradiation. For this study, four CdS/TiO_2_ samples were synthesized with different weight ratios of CdS in TiO_2_: S1 (9%), S2 (23%), S3 (45%) and S4 (74%). All CdS/TiO_2_ nanocomposites had higher photocatalytic activity than the TiO_2_ and CdS samples. The CO was the majoritarian product of this reduction reaction. The presence of CdS greatly improved the photocatalytic efficiency of the TiO_2_, and the best performance was achieved using the composite CdS/TiO_2_ S3 (45%), which was the optimal ratio of CdS/TiO_2_ for CO_2_ photoreduction. The enhancement used the CdS/TiO_2_ S3 (45%) composite, due to the large specific surface area and low crystal size of this sample.

As in the works reported above, the ratio of CdS in the composites was crucial to the photocatalytic performance of the composites. In this case, the authors also reported that the crystal size and specific surface area were the parameters that influenced the performance of these composites for the CO_2_ reduction. Therefore, a certain amount of CdS can enhance the TiO_2_ photocatalytic activity, and the porous structure of this CdS/TiO_2_ composite can have reacting sites for electrons transference to the reactant and avoid the recombination of the e^−^/h^+^ [47].

### 2.5. MoS_2_/TiO_2_

Molybdenum disulfide, MoS_2_, is a typical representative of two-dimensional (2D) transition metal chalcogenides (transition metal dichalcogenides—TMDs) [77]. Any one layer of MoS_2_ contains three atomic layers (S–Mo–S) stacked together [78]. This material is used as a substitute for noble metal co-catalysts, due to its properties, such as high activity, low cost, excellent chemical stability and abundance, and the band gap being around 1.3 to 1.9 eV [79]. Coupling TiO_2_ with MoS_2_ [49,80] leads to the creation of a heterojunction structure, which can speed up the electron transfer and reduce the photogenerated electrons and holes recombination. This heterojunction composite was recently studied for application on photocatalytic systems. MoS_2_/TiO_2_ composites were widely investigated as photocatalysts for photocatalytic degradation, hydrogen evolution, and CO_2_ reduction; with this combination, the reduction of the electron/hole recombination should be possible, and also the presence of MoS_2_ provides large catalytically active sites for photocatalytic progress [79]. CO_2_ photocatalytic reduction using MoS_2_/TiO_2_ heterojunction photocatalysts is tabulated in Table 6.

Peng-yao Jia et al. in 2019 [49] synthesized the MoS_2_/TiO_2_ heterojunction composites with different mass ratios of MoS_2_, 0, 5, 10 and 15 wt.%, obtaining the samples of TiO_2_, 5 wt.% MoS_2_/TiO_2_, 10 wt.% MoS_2_/TiO_2_, and 15 wt.% MoS_2_/TiO_2_, respectively. The obtained results showed that the 10% MoS_2_/TiO_2_ sample had higher photocatalytic activity for the photoreduction of CO_2_, obtaining higher amounts of CO and CH_4_ than the other composite materials. The achieved yields of CH_4_ and CO on the 10% MoS_2_/TiO_2_ heterojunction photocatalyst were approximately 5 times and 16 times higher than for pure TiO_2_ (P25). This result can be explained due to the lower band gap energy of this material, and also this material showed the lowest e^−^/h^+^ recombination by PL characterization; the photocurrent characterization indicates that this sample had more enhancement in e^−^ and h^+^ separation. The authors proposed a type-II heterojunction mechanism for this sample as can be seen in Table 6. Once the CB edge potential of MoS_2_ (−0.93 V) is more negative than that of TiO_2_ (−0.55 V), the migration of e^−^ from the surface of MoS_2_ to accumulate in the TiO_2_ is possible, due to the contact in the interface [49].

Xu et al. [50] described the preparation of 1D/2D TiO_2_/MoS_2_ nanostructured photocatalysts for increased photocatalytic CO_2_ reduction. The authors prepared different TiO_2_/MoS_2_ samples, with 1%, 5%, 7.5%, 10%, 15% and 25% of MoS_2_, labelled as TMx, where T and M are TiO_2_ and MoS_2_, respectively, and x denotes the mol.% of MoS_2_ to TiO_2_.

They observed that the formation rate of CH_4_ and CH_3_OH was markedly increased with higher MoS_2_ loading. The maximum value for the CH_4_ and CH_3_OH yields was reached for the TM10 sample. On the other side, only CH_3_OH was observed as the product in the presence of pure TiO_2_.

The authors concluded that the enhanced photocatalytic efficiency of TM10 is attributed to the increased light absorption, implying that more optical energy is absorbed after hybridization; the increased specific surface area nominates a higher amount of accessible reactive sites between TiO_2_/MoS_2_ and CO_2_ molecules; there is a higher CO_2_ adsorption capacity since CO_2_ adsorption is the beginning step for the next reduction processes; and there is enhanced charge separation, owing to the presence of MoS_2_ nanosheets as a cophotocatalyst [50].

In the TiO_2_/MoS_2_ samples, the photoinduced e^−^ in TiO_2_ transfers to MoS_2_, reaching more efficient electron–hole separation. Thus, the number of catalytically active e^−^ is significantly improved over pure TiO_2_, favoring the 8 e^−^ reaction for producing CH_4_. For that reason, the TMx showed higher photocatalytic CO_2_ reduction activity and better selectivity of CH_4_ than the pure TiO_2_ photocatalyst. However, the next increasing amount of MoS_2_ resulted in a decrease in the photocatalytic activity (e.g., TM25 and TM15), probably because of the severe charge carrier recombination and the shielding effects toward light absorption or the transfer of electrons, owing to the presence of a high amount of MoS_2_.

In this study, the author also analyzed the stability of the TM10 catalyst in consecutive reutilizations and concluded that TM10 is stable without loss of photoactivity for four cycles.

Test with an isotope tracer confirmed that the products of CO_2_ photocatalytic reduction solely originated from the CO_2_ source. The DFT calculation demonstrated that TiO_2_ has a higher work function than MoS_2_, resulting in electrons transfer from MoS_2_ to TiO_2_ upon their contact, which supports the charge carrier separation of upon photoexcitation as MoS_2_ acts as a cophotocatalyst. Moreover, the hybridization with MoS_2_ increases light harvesting and enhances the CO_2_ adsorption of TiO_2_, further contributing to the superior photocatalytic efficiency of the TiO_2_/MoS_2_ hybrid [50].

Tu et al. [51] described the preparation of two-dimensional MoS_2_–TiO_2_ hybrid nanojunctions, for the CO_2_ photocatalytic reduction to CH_3_OH. They prepared MoS_2_/TiO_2_ photocatalysts with 3, 2, 1, and 0.5 wt.% contents of MoS_2_.

All samples proved photocatalytic activity for CO_2_ photoreduction into CH_3_OH, the 0.5 wt.% MoS_2_/TiO_2_ sample being the one with the best photocatalytic performance for this reaction. Using this sample, CH_3_OH production was almost three times higher than using pure TiO_2_. However, for the samples with higher MoS_2_ content (1, 2, and 3 wt.%) a gradual decrease in the photocatalytic activity was obtained. This occurs due to the fact that the photons in the photocatalytic system are absorbed by the excess of black MoS_2_ nanosheets, and probably decrease the light intensity through shielding the light reached on the TiO_2_ surface (i.e., “shielding effect”).

It was found that the two-dimensional MoS_2_/TiO_2_ hybrid composites presented high photocatalytic activity for CO_2_ photoreduction. They concluded that loaded MoS_2_ nanosheets minimize the charge carrier recombination and enhance the conversion performance of the CO_2_ photoreduction into CH_3_OH due to the e^−^ transfer from TiO_2_ to MoS_2_ [51].

### 2.6. Other Semiconductors

In this section, studies on the CO_2_ photoreduction using composite materials with heterojunction with TiO_2_ not so often used until now are shown. CO_2_ photocatalytic reduction using GaP/TiO_2_, CaTiO_3_/TiO_2_ and FeTiO_3_/TiO_2_ heterojunction photocatalysts are tabulated in Table 7.

#### 2.6.1. GaP/TiO_2_

Gallium phosphide, GaP, is a semiconductor material with an indirect band gap of 2.3 eV, insoluble in water. This semiconductor is not often used as a photocatalyst due to the low oxidizing power of its VB; however, the conduction band (CB) position allows the CO_2_ reduction once it is 1.26 V more negative than that of CO_2_/CH_4_ (*E*^0^ = −0.24 V) and 0.97 V more negative than CO_2_/CO (*E*^0^ = −0.53 V), as can be seen in the reactions from Table 1.

In 1978, Halmann [81] used GaP for the photoelectrochemical reduction of CO_2_. In this case, GaP was used in the liquid junction of solar cells, and the obtained products were formic acid, formaldehyde and methanol [52,81]. Furthermore, recently, Barton et al. also used GaP and found that a highly selective CO_2_ photoreduction to CH_3_OH occurred when a GaP electrode with pyridine was used. In this case, pyridine served as a cocatalyst [52,82]. Regardless of the fact that electrons from the GaP conduction band can reduce CO_2_ to methane, we need to look at the oxidation reaction as well. For example, often water or water vapor is chosen as the hole trap (oxidation step). In this case, water cannot be used as a hole trap because the GaP valence band (*E*^0^ = 0.80 V) does not have sufficient potential for water oxidation (*E*^0^ = 0.82 V). Therefore, the pristine GaP cannot be used for the CO_2_ photocatalytic reduction.

In line with this, Giuseppe Marcì et al. [52], for the first time, evaluated the GaP/TiO_2_ composites for the photocatalytic reduction of carbon dioxide. The suitable position of VB and CB of the semiconductors not only allows for heterojunction photocatalysts (GaP/TiO_2_) to have efficient electron–hole separation, but also enables both H_2_O oxidation and CO_2_ reduction.

Giuseppe Marcì et al. [52] reported GaP/TiO_2_ photocatalysts with significant efficiency during the photocatalytic reduction of carbon dioxide to the formation of methane. The researchers concluded that decreasing the mass ratio of the GaP:TiO_2_ enhances the photoactivity of the photocatalyst, and the highest efficiency was observed in the presence of photocatalysts with a 1:10 mass ratio. The photocatalytic effectiveness of the photocatalysts was connected with the band structures of the semiconductors and also with the efficient electron–hole transfer between GaP and TiO_2_ in the heterojunction photocatalysts.

#### 2.6.2. FeTiO_3_/TiO_2_

Another interesting alternative to improve the TiO_2_ photocatalytic reduction of CO_2_ is the combination with ternary oxides, such as ilmenite (FeTiO_3_) and perovskite (CaTiO_3_); these heterojunction composite materials have not been studied very much for the photoreduction of CO_2_. However, the works reported by Truong et al. [53] and Lin et al. [54] showed that FeTiO_3_/TiO_2_ and CaTiO_3_/TiO_2_, respectively, are promising materials for this photocatalytic reaction.

Ilmenite (FeTiO_3_) is a semiconductor material with a band gap energy between 2.59 and 2.90 eV. This is one of the most abundant minerals used as raw material for the production of TiO_2_ and Ti. FeTiO_3_ has been studied by several researchers, due to its optic, semiconductive and magnetic properties, low-cost and high abundancy (as natural ilmenite), being an alternative semiconductor for photoactivated processes [83,84,85,86,87,88]. This semiconductor has been used for the formation of hetero-interfaces, with other different semiconductors, such as p–n junctions and Schottky contacts for effective carrier separation [89]. Recently, it was reported the high efficiency of FeTiO_3_ as a photocatalyst for hydrogen production [83]. Furthermore, several works have been reported with the preparation and utilization of FeTiO_3_-TiO_2_ composites as photocatalysts for the degradation of organic pollutants, showing that this combination improves the photocatalytic activity [84,85,90,91].

Truong et al. [53] showed the photocatalytic reduction of CO_2_ using the FeTiO_3_/TiO_2_ photocatalyst. The authors reported the preparation of a heterojunction sample of FeTiO_3_/TiO_2_ with various Fe/Ti mole ratios of 70%, 50%, 20%, and 10% [53]. All FeTiO_3_/TiO_2_ composites had a significantly higher photoactivity for the carbon dioxide reduction under both radiation sources (UV–Vis and visible light) in comparison with the TiO_2_ and P25 samples. This can be explained due to the heterojunction effect between the two semiconductors, and also the higher activity in the visible light due to the combination of TiO_2_ with FeTiO_3_. In the FeTiO_3_/TiO_2_ composites, the e^−^ in the valence band of TiO_2_ transfer to FeTiO_3_ VB, while the h^+^ are subsequently created in TiO_2_ CB. Furthermore, the e^−^/h^+^ are photogenerated, owing to the narrow bandgap of FeTiO_3_ [53,91]. The obtained results showed that 20% FeTiO_3_/TiO_2_ sample had the best photocatalytic activity. In cases with a higher amount of FeTiO_3_ (50% and 70%), the CH_3_OH production decreased. This was explained by the smaller surface area, and also the high metal amount in FeTiO_3_/TiO_2_, which can represent recombination centers, resulting in reduced photocatalytic efficiency. The enhanced photoactivity along with an increasing amount of FeTiO_3_ is reasonable, due to the higher number of active sites for the carbonate species reduction. The optimal FeTiO_3_ amount for the highest photocatalytic efficiency is 20 wt.%. The authors concluded with this study that the unique band structure, the heterojunction effect of two materials, and the FeTiO_3_ narrow bandgap were responsible for the significant photocatalytic effectiveness on selective CH_3_OH production during CO_2_ photoreduction [53].

#### 2.6.3. CaTiO_3_/TiO_2_

CaTiO_3_ is a titanium-based perovskite-type oxide, and an n-type semiconductor with a large band gap between 3.0 and 3.5 eV. This is an alkaline earth metal titanate that is non-toxic, with chemical stability, optical properties, a low cost and an eco-friendly nature. Currently, it is being used for several applications, such as electronic devices, photocatalytic degradation of dyes, water splitting for H_2_ production and CO_2_ reduction [92]. CaTiO_3_ has been studied for the preparation of heterostructured photocatalysts systems to improve their photocatalytic activity, in order to promote separation and photogenerated charge carrier transportation, also leading to the improvement in their visible light response. For example, coupling CaTiO_3_ with TiO_2_ was studied for organic pollutants’ photodegradation [93]; however, for the CO_2_ reduction, only one study was reported to date.

Lin et al. [54] synthesized four CaTiO_3_/TiO_2_ composite samples with different amounts of TiO_2_, 0.4, 0.3, 0.2 and 0.1 g, obtaining samples named 8.6%CaTiO_3_/TiO_2_, 13.4%CaTiO_3_/TiO_2_, 24.2%CaTiO_3_/TiO_2_ and 66.7%CaTiO_3_/TiO_2_, respectively, with the weight contents of CaTiO_3_ obtained by XRD quantification.

The activity of these four CaTiO_3_/TiO_2_ had higher photoactivity for the CO_2_ photoreduction than the TiO_2_ and P25 samples. The best sample for this reduction reaction was 13.4%CaTiO_3_/TiO_2_, being that the photocatalytic activity of this sample was six times higher than the TiO_2_. The results also showed that the CaTiO_3_ and TiO_2_ ratio influenced the photocatalytic activity efficiency for the CO_2_ photoreduction, i.e., the increase in CaTiO_3_ content above 13.4%CaTiO_3_/TiO_2_ decreased the CO evolution. The authors reported that when an excess of CaTiO_3_ content was introduced, a nanocubic morphology was obtained instead of a nanosheets morphology, and also the specific surface area decreased, being the reason for the decrease in the CO production. The authors concluded that the enhancement in the CaTiO_3_/TiO_2_ composites catalytic activity for CO_2_ photoreduction was attributed to the similar crystal structures and the matched band structures of the CaTiO_3_/TiO_2_ heterojunction photocatalysts that simplified the photogenerated electron–hole separation, as well as the increased surface basicity of the CaTiO_3_/TiO_2_ samples that provided more abundant active sites for adsorption of CO_2_ and, therefore, increased the photoreduction CO_2_ [54].

### 2.7. Semiconductor-Covalent Organic Framework Z-Scheme Heterojunctions

The integration of covalent organic frameworks (COFs) with inorganic materials gives possibilities to develop new type of composite materials [94]. These materials have high surface areas and novel functionalities relevant to photocatalysis, chemical adsorption, and magnetic resonance imaging. The disadvantages of these materials associated with challenging, multi-step synthesis were overcome by Zhu et al. [94], who reported a one-pot synthesis approach, using a wide range of metal oxides to catalyze the synthesis of highly crystalline and porous COFs.

A series of COF semiconductor Z-scheme photocatalysts integrating semiconductors (TiO_2_, Bi_2_WO_6_ and α-Fe_2_O_3_) with COFs (COF-316/318) were synthesized and characterized by Zhang et al. [95]. Prepared photocatalysts showed high photocatalytic CO_2_ conversion to CO efficiency, with H_2_O as an electron donor in the gas–solid CO_2_ reduction without additional photosensitizers and sacrificial agents. This is the first report of a covalent-bonded COF-inorganic semiconductor Z-scheme applied for artificial photosynthesis. The COF-318-TiO_2_ Z-scheme heterojunction photocatalyst showed the highest CO production rate, which was about six times higher than the pure COF-318, and TiO_2_ was also much higher than the physical mixture composites. Experiment studies and density functional theory (DFT) confirmed the efficient electron transfer from semiconductors to COFs by covalent coupling, resulting in the electrons being accumulated in cyano/pyridine of COF for the reduction of CO_2_ and positive holes remaining in the semiconductor for the oxidation of H_2_O. This work found a new method to create a covalent bond linked organic–inorganic Z-scheme heterojunction and showed a new perspective in the field of photocatalysis.

## 3. Final Conclusions

Nowadays, energy depletion and environmental pollution is one of the most discussed topics. The photocatalytic reduction of carbon dioxide into valuable and clean fuels can be one of the sustainable solutions to reduce carbon dioxide emissions. Although photocatalytic CO_2_ reduction has received unprecedented attention from scientists worldwide, its widespread use is limited due to the low selectivity, stability and especially the low efficiency of the photocatalytic system. The most studied photocatalyst in recent years is TiO_2_ because it is cheap, non-toxic and environmentally friendly. Unfortunately, TiO_2_ has some limitations, such as its activation, especially in the UV region, or the rapid recombination of generated electrons and holes. These imperfections can be tuned by doping TiO_2_ with metals or non-metals or by creating TiO_2_ heterojunction photocatalysts with other semiconductors.

In this review, TiO_2_ heterojunction photocatalysts were discussed to further increase the photocatalytic efficiency of TiO_2_ photocatalysts. In the last few years, several studies have been published on the preparation of TiO_2_ heterojunction photocatalysts suitable for photocatalytic CO_2_ reduction. Using these materials with the heterojunction, it was possible to improve the catalytic activity for the photoreduction of CO_2_, due to the efficient electron transference in the interface, supporting the separation of the e^−^/h^+^ pairs and consequently reducing the e^−^/h^+^ recombination. In addition, the activity in the visible light range was improved because it was possible to utilize sunlight more effectively; there was higher adsorption of CO_2_ due to the highly specific surface area; and there was an increase in selectivity specific CO_2_ photoreduction products due to the contribution of the cocatalysts. In the case of heterojunction photocatalysts, there is always an optimal amount or ratio of semiconductors used. The use of the metal as a dopant TiO_2_, which then forms a heterojunction with C_3_N_4_, also proved to be very advantageous.

Further research in this area should focus on the following aspects:To create heterojunction photocatalysts, it is essential to find materials that have the appropriate band structure for redox reactions, are active in the visible light region, and are stable.Efforts are underway to develop not too complex, efficient and effective methods for preparing heterojunction photocatalysts that could be produced in larger quantities. The most appropriate physicochemical properties of each semiconductor, such as the appropriate morphology, crystallite size, phase composition, etc., should be considered when developing preparation methods.The migration pathways of photogenerated electron–hole pairs need to be thoroughly studied. Heterojunction photocatalysts can have different arrangements (e.g., heterojunction type II or Z-scheme heterojunction) and, thus, different migration pathways for electron–hole separation, which need to be thoroughly studied and confirmed using advanced characterization techniques.To better understand the mechanism of migration pathways, knowledge from modeling methods or theoretical calculations should be used.

We hope that this review will encourage new approaches to the preparation of heterojunction photocatalysts, help optimize existing photocatalysts and create new efficient heterojunctions to achieve the higher efficiencies that are necessary for practical applications.

## Figures and Tables

**Figure 1 materials-15-00967-f001:**
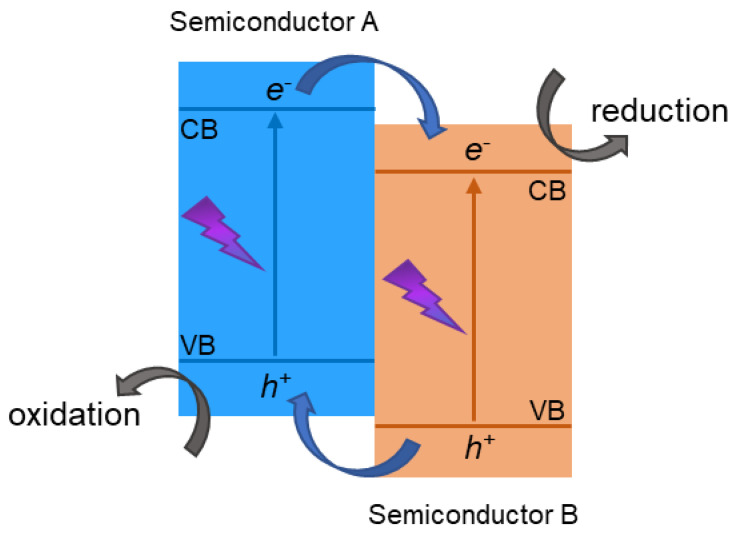
Schematic illustration of the electron–hole separation on an example of heterojunction photocatalyst type-II. Adapted according to refs. [18,19].

**Figure 2 materials-15-00967-f002:**
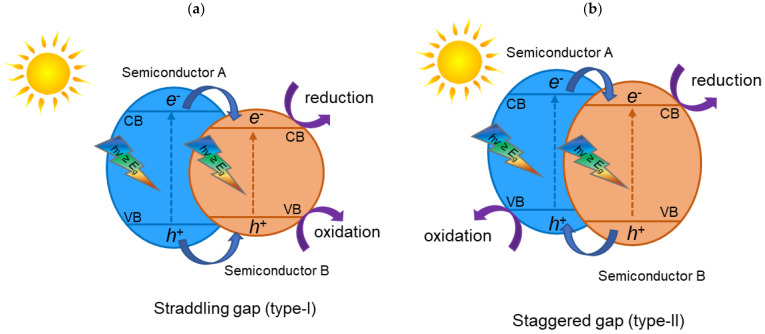
Schematic illustration of the three different types of separation of electron–hole pairs in the case of conventional light-responsive heterojunction photocatalysts: (**a**) type-I, (**b**) type-II, and (**c**) type-III heterojunctions. Adapted according to Refs. [18,19].

**Figure 3 materials-15-00967-f003:**
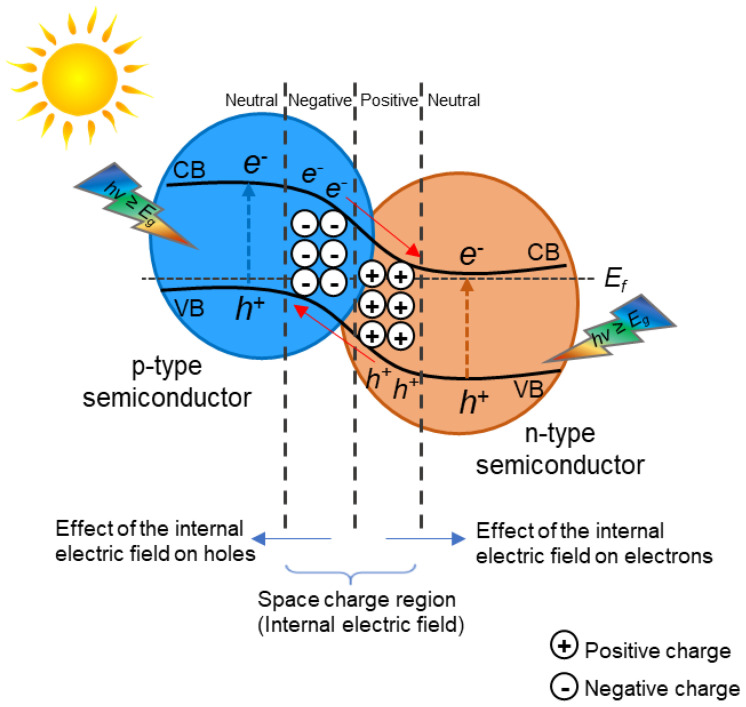
Schematic illustration of the electron–hole separation under the influence of the internal electric field of a p–n heterojunction photocatalyst under light irradiation. Adapted according to Refs. [18,19].

**Figure 4 materials-15-00967-f004:**
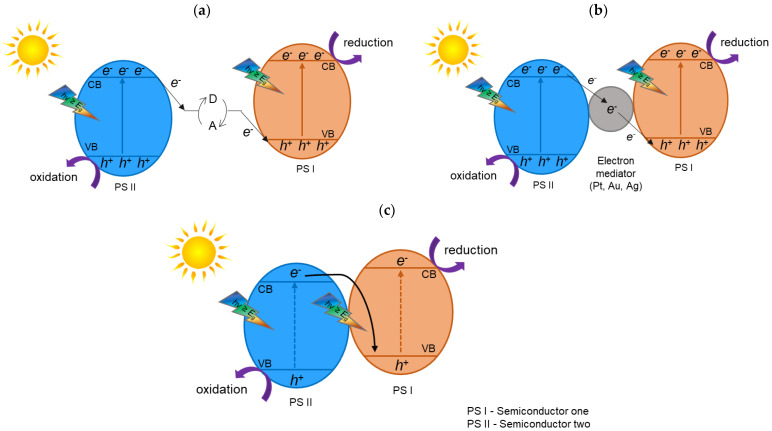
Schematic representation of (**a**) electron–hole separation on the conventional Z-scheme photocatalytic system; (**b**) the electron–hole separation on all-solid-state Z-scheme photocatalysts; and (**c**) electron–hole separation on a direct Z-scheme heterojunction photocatalyst. Adapted according to Refs. [18,19].

**Table 1 materials-15-00967-t001:** Main products of CO_2_ reduction and the corresponding potential (pH = 7).

Reaction	*E°* (V vs. NHE)	Product	Reference
$H_{2}O+2e^{-}\to 2{\mathrm{OH}}^{-}+{\;H}_{2}$	−0.41	Hydrogen	[8]
${\mathrm{CO}}_{2}+e^{-}\to {\mathrm{CO}}_{2}^{\cdot -}$	−1.90	$•{\mathrm{CO}}_{2}^{-}$ anion radical	[9]
$2{\mathrm{CO}}_{2}+2H^{+}+2e^{-}\to H_{2}C_{2}O_{4}$	−0.87	Oxalate	[8]
${\mathrm{CO}}_{2}+2H^{+}+2e^{-}\to \mathrm{HCOOH}$	−0.61	Formic acid	[10]
${\mathrm{CO}}_{2}+2H^{+}+2e^{-}\to \mathrm{CO}+{\;H}_{2}O$	−0.53	Carbon monoxide	[9,10]
${\mathrm{CO}}_{2}+4H^{+}+4e^{-}\to \mathrm{HCHO}+{\;H}_{2}O$	−0.48	Formaldehyde	[8,9,10]
${\mathrm{CO}}_{2}+6H^{+}+6e^{-}\to {\mathrm{CH}}_{3}\mathrm{OH}+{\;H}_{2}O$	−0.38	Methanol	[9,10]
$2{\mathrm{CO}}_{2}+12H^{+}+12e^{-}\to C_{2}H_{5}\mathrm{OH}+3H_{2}O$	−0.33	Ethanol	[8]
$2{\mathrm{CO}}_{2}+14H^{+}+14e^{-}\to C_{2}H_{6}+4H_{2}O$	−0.27	Ethane	[8]
${\mathrm{CO}}_{2}+8H^{+}+8e^{-}\to {\mathrm{CH}}_{4}+2H_{2}O$	−0.24	Methane	[8,9,10]

**Table 2 materials-15-00967-t002:** CO_2_ photoreduction using g-C_3_N_4_/TiO_2_ heterojunction photocatalysts.

Photocatalysts		CO_2_ Photoreduction Condition			Yield of Products	Type of Heterojunction		Ref.
Type	Prepared	Reaction Mixture	Light Source	Conditions				
TiO_2−x_/g-C_3_N_4_	Solid state synthesis	CO_2_ (99.999%), 5 mL of solution containing 4 mL of methyl cyanide (MeCN) solvent, 1 mL of triethanolamine (TEOA), bipyridine (bpy) (10 mmol L^−1^) and 25 μL of 20 mmol L^−1^ CoCl_2_ aqueous solution	300 W xenon lamp	43 mL quartz vessel with a rubber septum; 25 °C; circulation cooling system. Photocatalyst concentration in 1 g L^−1^	CO = 77.8 μmol g^−1^ h^−1^	Type-II 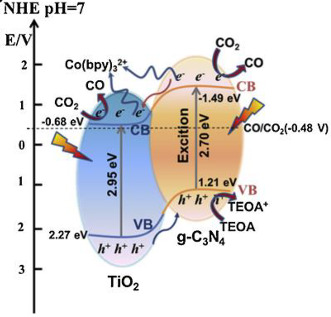	*	[25]
(0.3/1)TiO_2_/g-C_3_N_4_	Simple mechanical mixing of pure g-C_3_N_4_ and commercial TiO_2_ Evonik P25	CO_2_ with a certified maximum of hydrocarbons less than 1 ppm (SIAD Technical Gases, CZ)	8 W Hg lamp	Cylindrical stirred batch reactor, with internal volume of 355 cm^3^ Photocatalyst concentration in 0.28 g L^−1^	CH_4_ = 70 μmol g_cat._^−1^ CO = 23 μmol g_cat._^−1^ after 8 h	Type-II 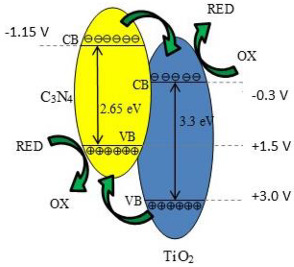	‡	[26]
TiO_2_@g-C_3_N_4_-20%	Stirring method	CO_2_ and 50 mL 0.08 mol L^−1^ NaHCO_3_ solution	300 W Xe lamp with a 420 nm optical filter	quartz glass tube with a volume of 60 mL Photocatalyst concentration in 1 g L^−1^	CH_3_OH ~50 μmol g_cat_^−1^ after 4 h	Type-II(see Ref. [27])	-	[27]
HCNS@TiO_2_	Templating method combined with the subsequent kinetically-controlled coating process	CO_2_ (high purity) and H_2_O (400 mL)	Visible-light (300 W Xenon lamp)	cylindrical Pyrex glass photoreactor with 500 mL of volume Photocatalyst concentration in 1 g L^−1^	CH_3_OH = 52.1 μmol g_cat_^−1^ CH_4_ = 21.3 μmol g_cat_^−1^ after 6 h	Type-II 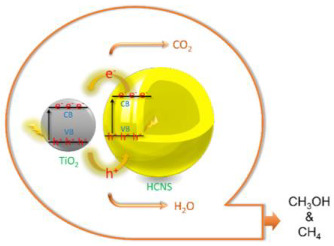	†	[28]
70:30 g-C_3_N_4_-N-TiO_2_	Hydrothermal method and thermal treatment (in situ method)	Deionized H_2_O + CO_2_ (99.999%)	300 W Xe arc lamp Intensity 100 mW/cm^2^	780 mL gas-closed circulation Teflon system Photocatalyst concentration in 0.13 g L^−1^	CO = 14.73 μmol after 12 h	Type-II 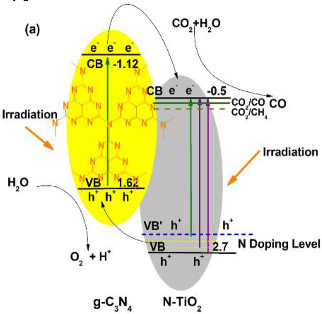	¥	[29]
Nb-TiO_2_/g-C_3_N_4_	Solid state synthesis	CO_2_ (99.99%) flow rate 20 mL/min; water vapor was used as hole scavenger	Two 30 W white bulbs	continuous gas system with a reactor (40 mL) located in the center of a dark cover cask using as a reaction chamber (24 L) Photocatalyst concentration in 2.5 g L^−1^	CO = 420 μmol g^−1^ h^−1^ HCOOH = 698 μmol g^−1^ h^−1^ CH_4_ = 562 μmol g^−1^ h^−1^	Z-scheme 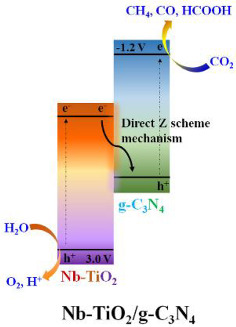	§	[30]
8 mass % g-C_3_N_4_/Ag-TiO_2_	Solvent evaporation followed by calcination	CO_2_ flow rate 3 mL/min; water vapor was used as hole scavenger	300 W xenon lamp	70 mL cylindrical photoreactor Photocatalyst concentration in 0.7 g L^−1^	CH_4_ = 28.0 μmol g^−1^ CO = 19.0 μmol g^−1^ after 3 h	Type-II 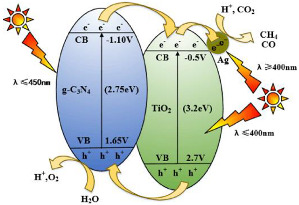	ƗƗ	[31]
Phosphate–oxygen (P–O) bridged TiO_2_/g-C_3_N_4_	Impregnation-solid state synthesis	CO_2_ + 3 mL H_2_O; water vapor was used as a hole scavenger	300 W xenon lamp	cylindrical steel reactor (volume of 100 mL and area of 3.5 cm^2^) Photocatalyst concentration in 2 g L^−1^	CH_4_ = 40 μmol g^−1^ h^−1^ CO = 15 μmol g^−1^ h^−1^	Z-scheme 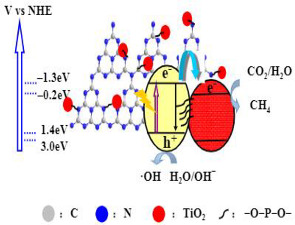	I	[32]
(Au, C_3_N_4_)/TiO_2_	Immersing (or dipping) method	CO_2_ + 5 mL H_2_O	300 W Xenon arc lamp	100 mL sealed steel container with cooling water Photocatalyst: Two pieces of samples (0.5 cm^2^/sample	CO = 0.138 µmol cm^−2^h^−1^ CH_4_ = 0.032 µmol cm^−2^h^−1^	Z-scheme 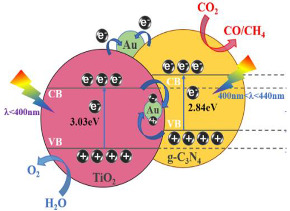	II	[33]

* Reprinted from [25], Copyright (2019), with permission from Elsevier. ‡ Reprinted with permission from [26]. Copyright 2016 American Chemical Society. † Reprinted from [28], Copyright (2020), with permission from Elsevier. ¥ Reprinted from [29], Copyright (2014), with permission from Elsevier. § Reprinted from [30], Copyright (2019), with permission from Elsevier. ƗƗ Reprinted from [31], Copyright (2017), with permission from Elsevier. I Reprinted from [32], Copyright (2017), with permission from Elsevier. II Reprinted from [33], Copyright (2019), with permission from Elsevier.

**Table 3 materials-15-00967-t003:** CO_2_ photoreduction using CeO_2_/TiO_2_ heterojunction photocatalysts.

Photocatalysts		CO_2_ Photoreduction Condition			Yield of Products	Type of Heterojunction		Ref.
Type	Prepared	Reaction Mixture	Light Source	Conditions				
Mes-CeTi-1.0	Template method using a nanocasting route	CO_2_ + H_2_O	Xe arc lamp 300 W	stainless steel reactor (volume of 1500 mL) Photocatalyst concentration in 0.07 g L^−1^	CH_4_ = 11.5 mmol g_cat_^−1^ CO = ~70 mmol g_cat_^−1^ after 325 min	-	-	[34]
CeO_2_-TiO_2_	Stirring method and calcination method	CO_2_ and 300 mL of 0.1 mol L^−1^ NaOH solution (for 30 min before irradiation) During irradiation CO_2_ was continuously bubbled	Visible light—500 W Xenon lamp, and 2 mol L^−1^ sodium nitrite solution (to remove UV light)	Pyrex glass reactor (500 mL) Photocatalyst concentration in 1 g L^−1^	CH_3_OH = 18.6 μmol g_cat_^−1^ after 6 h	Type-II 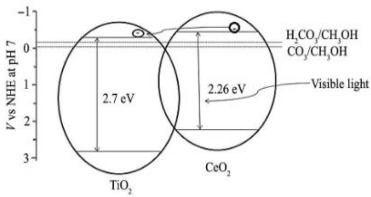	*	[35]
CeO_2_/TiO_2_-4	Gas bubbling-assisted membrane precipitation (GBMP) method	CO_2_ and H_2_O	300 W Xe lamp and an optical filter with the absorbed light wavelength of <420 nm	Glass reactor (basal diameter of 4 cm) Photocatalyst amount 0.1 g	CO = 2.06 μmol after 6 h	Type-II 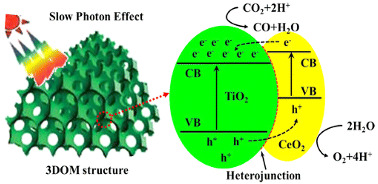	‡	[36]
CeO_2_/TiO_2_ (R-TiCe_0.1_)	Hydrothermal method	CO_2_ and H_2_O (Gaseous CO_2_ of 8 kPa was in site produced by the reaction of NaHCO_3_ with a H_2_SO_4_ solution (0.5 M).)	500 W Xenon lamp	reactor connected with mechanical vacuum pump Photocatalyst amount 10 mg	CO = 61.9 μmol g^−1^ CH_4_ = 23.5 μmol g^−1^ after 6 h	Type-II 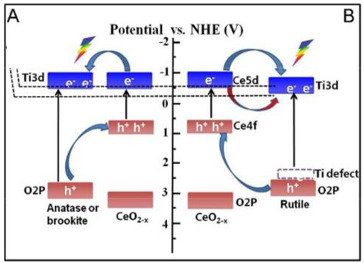	†	[37]
0.2CeO_2_/TiO_2_	One-pot hydrothermal method	CO_2_ and H_2_O (Gaseous CO_2_ of 8 kPa was produced in situ by the reaction of NaHCO_3_ with a H_2_SO_4_ solution (0.5 M).)	300 W Xenon lamp	reactor connected with mechanical vacuum pump Photocatalyst amount 10 mg	CO = 46.6 μmol g^−1^ CH_4_ = 30.2 μmol g^−1^ after 6 h	Type-II 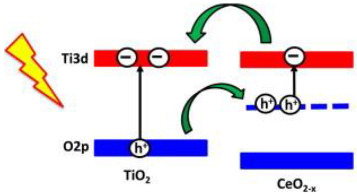	¥	[38]

* Reprinted from [35], Copyright (2015), with permission from Elsevier. ‡ Reprinted with permission from [36]. Copyright 2014 American Chemical Society. † Republished with permission of Royal Society of Chemistry, from [37] copyright 2016; permission conveyed through Copyright Clearance Center, Inc. ¥ Reprinted from [38], Copyright (2016), with permission from Elsevier.

**Table 4 materials-15-00967-t004:** CO_2_ photoreduction using CuO/TiO_2_ heterojunction photocatalysts.

Photocatalysts		CO_2_ Photoreduction Condition			Yield of Products	Type of Heterojunction		Ref.
Type	Prepared	Reaction Mixture	Light Source	Conditions				
CuO/TiO_2_(AB)	Impregnation method	pure CH_3_OH solution (30 mL), and pure CO_2_ gas	250 W Hg lamp intensity 3900 μW/cm^2^ at 365 nm	ideal mixing 50 mL quartz tube Photocatalyst concentration in 1 g L^−1^	HCOOCH_3_ = ~1800 μmol g_cat_^−1^ after 4 h	-	-	[41]
3 wt.% CuO/TiO_2_	Impregnation method	CO_2_ (Ultra high purity grade), 130 mL of 0.2 M KHCO_3_ and 0.1 M Na_2_SO_3_ aqueous solutions	500 W high pressure Hg lamp with a peak light intensity at 365 nm	quartz reactor Photocatalyst concentration in 2.77 g L^−1^	methanol = 12.5 μmol g^−1^ ethanol = 27.1 μmol g^−1^ after 6 h	--	-	[42]
1.0CuO-TiO_2_	Stirring method followed by calcination	CO_2_ (99.99% purity) and 30 mL of methanol	250 W high pressure mercury lamp with the radiation peak at about 365 nm	slurry reactor system Photocatalyst concentration in 1 g L^−1^	Methyl formate ~1600 μmol g^−1^ h^−1^	Z-scheme 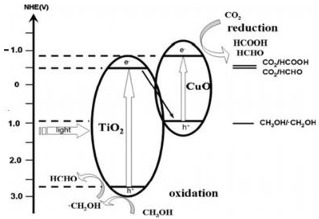	*	[39]
CuO loaded TiO_2_ nanotube	Hydrothermal method	CO_2_ (flow rate of 30 mL min^−1^) and ultrapure water, and NaHCO_3_ (0.1 M)	400 W high-pressure mercury lamp with a quartz filter	flow system with an inner-irradiation-type reaction vessel at ambient pressure Photocatalyst amount 0.5 g	100% CO_2_ conversion into CH_4_ and CH_3_OH after 2.5 h	Type-I 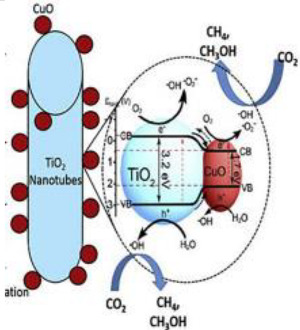	‡	[43]

* Reprinted from [39], Copyright (2011), with permission from Elsevier. ‡ Reprinted from [43], Copyright (2018), with permission from Elsevier.

**Table 5 materials-15-00967-t005:** CO_2_ photoreduction using CdS/TiO_2_ heterojunction photocatalysts.

Photocatalysts		CO_2_ Photoreduction Condition			Yield of Products	Type of Heterojunction		Ref.
Type	Prepared	Reaction Mixture	Light Source	Conditions				
TiO_2_/CdS-3	Conventional hydrothermal technique	Ar or CO_2_ (both 99.99%) for 1 h, and aqueous isopropanol solution (1.0 M, 100 mL)	450 W Xe arc lamp in combination with 320 nm or 420-nm-cutoff filters	airtight glass reactor (120 mL) with a quartz disc for light penetration Photocatalyst concentration in 1 g L^−1^	methane = ~18 µmol (after 10 h) CO = ~2.5 µmol (after 10 h) Under UV-vis irradiation	Type-II 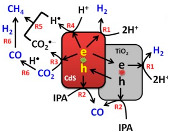	*	[44]
TiO_2_/CdS	Ionic layer adsorption and reaction (SILAR) method	CO_2_ and H_2_O vapor (from 84 mg of NaHCO_3_ and 0.3 mL of HCl solution (4 M))	300 W Xenon arc lamp	200 mL Pyrex reactor (purged with N_2_ gas) Photocatalyst: Film with 4 cm^2^	11.9 mmol h^−1^ m^−2^ for CH_4_ production	Z-scheme 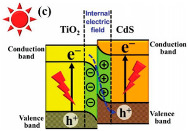	‡	[45]
CdS-TiO_2_-8	Hydrothermal method	CO_2_ and 10 mL cyclohexanol	250 W high pressure mercury lamp	batch slurry bed reactor with inner capacity of 50 mL Photocatalyst concentration in 2 g L^−1^	cyclohexyl formate = 20.2 µmol g_cat_^−1^h^−1^ cyclohexanone = 20 µmol g_cat_^−1^ h^−1^	Z-Scheme 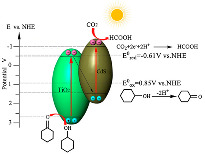	†	[46]
CdS-TiO_2_ S3 (45%)	Hydrothermal method	N_2_ and CO_2_	125 W Hg lamp (350–400 nm) For the visible light, the UV wavelengths <400 nm were removed using a sodium nitrite solution (2.0 M)	Pyrex reactor with an effective volume of 125 mL Photocatalyst concentration in 1.44 g L^−1^	Under UV-vis irradiation: CO = ~15.5 µmol g_cat_^−1^ CH_4_ = ~3.0 µmol g_cat_^−1^ after 8 h Under visible light irradiation: CO = ~10.3 µmol g_cat_^−1^ CH_4_ = ~1.5 µmol g_cat_^−1^ after 8 h	Type-II 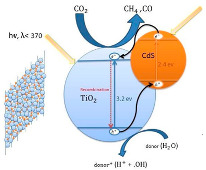	¥	[47]

* Reprinted from [44], Copyright (2016), with permission from Elsevier. ‡ Reprinted from [45], © 2022 WILEY-VCH Verlag GmbH & Co. KGaA, Weinheim. † Reprinted from [46], Copyright (2014), with permission from Elsevier. ¥ Reprinted from [47], Copyright (2014), with permission from Elsevier.

**Table 6 materials-15-00967-t006:** CO_2_ photoreduction using MoS_2_/TiO_2_ heterojunction photocatalysts.

Photocatalysts		CO_2_ Photoreduction Condition			Yield of Products	Type of Heterojunction		Ref.
Type	Prepared	Reaction Mixture	Light Source	Conditions				
10% MoS_2_/TiO_2_	Calcined at 300 °C for 4 h with argon shielding gas	100 mL deionized H_2_O which was preprocessed for 30 min with CO_2_ (99.99%) of 100 kPa	Xe-arc lamp 300 W acting	500 cm^3^ cylindrical reactor Photocatalyst concentration in 0.5 g L^−1^	CO = 268.97 μmol g_cat_^−1^ CH_4_ = 49.93 μmol g_cat_^−1^ after 6 h	Type-II 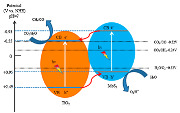	*	[49]
10% MoS_2_/TiO_2_	In situ growing MoS_2_ nanosheets onto TiO_2_ nanofibers by hydrothermal method	CO_2_ and H_2_O vapor were in situ generated by the reaction of NaHCO_3_ (0.12 g) and H_2_SO_4_ aqueous solution (0.25 mL, 2 M)	350 W Xe lamp	200 mL homemade Pyrex reactor Photocatalyst concentration in 0.25 g L^−1^	CH_4_ = 2.86 μmol g^−1^ h^−1^ CH_3_OH = 2.55 μmol g^−1^ h^−1^	Type-II 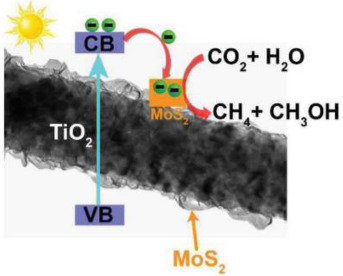	‡	[50]
0.5 wt% MoS_2_/TiO_2_	Hydrothermal method	200 mL of 1 M NaHCO_3_ solution and pure CO_2_	300 W Xenon arc lamp.	airtight quartz glass reactor Photocatalyst concentration in 0.5 g L^−1^	CH_3_OH = 10.6 μmol g^−1^ h^−1^	-		[51]

* Reprinted from [49], Copyright (2019), with permission from Elsevier. ‡ Reprinted from [50], © 2022 WILEY-VCH Verlag GmbH & Co. KGaA, Weinheim.

**Table 7 materials-15-00967-t007:** CO_2_ photoreduction using GaP/TiO_2_, CaTiO_3_/TiO_2_ and FeTiO_3_/TiO_2_ heterojunction photocatalysts.

Photocatalysts		CO_2_ Photoreduction Condition			Yield of Products	Type of Heterojunction		Ref.
Type	Prepared	Reaction Mixture	Light Source	Conditions				
1:10 GaP/TiO_2_	Mechanically milling of Commercial TiO_2_ Evonik P25 and GaP Aldrich powders	CO_2_ and water	1500 W high pressure Xe lamp	gas–solid Pyrex batch photoreactor of cylindrical shape (V = 100 mL, Φ = 94 mm, height = 15 mm) Photocatalyst concentration in 3 g L^−1^	CH_4_ = 118.18 μM g^−1^ after 10 h	Z-scheme 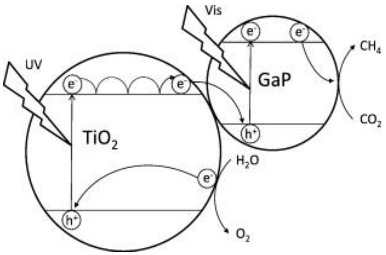	*	[52]
20% FeTiO_3_/TiO_2_	Hydrothermal method	30 mL distilled water containing sodium bicarbonate (NaHCO_3_, 0.08 M)	500 W high-pressure Xe lamp. A Pyrex glass tube cut off light with λ < 300 nm and a 2 M NaNO_2_ solution was applied to cut off λ < 400 nm	quartz reaction vessel, connected to a gas chromatograph. Photocatalyst concentration in 1.7 g L^−1^	CH_3_OH = 0.462 μmol g^−1^ h^−1^ under UV-vis irradiation and CH_3_OH = 0.432 μmol g^−1^ h^−1^ under visible light irradiation.	-	-	[53]
13.4% CaTiO_3_/TiO_2_	In situ hydrothermal method	CO_2_ and water	300 W Xe lamp	Quartz tube reactor, with 43 mL volume Photocatalyst concentration in 0.23 g L^−1^	CO = 11.72 μmol g^−1^ h^−1^	Z-scheme 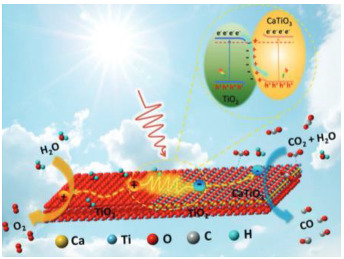	‡	[54]

* Reprinted from [52], Copyright (2014), with permission from Elsevier. ‡ Republished with permission of Royal Society of Chemistry, from [54] copyright 2019; permission conveyed through Copyright Clearance Center, Inc.

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
