# Peer review of "Photocatalytic Reduction of Carbon Dioxide on TiO2 Heterojunction Photocatalysts—A Review"

_materials, 2022, doi:10.3390/ma15030967_

Round 1

Reviewer 1 Report

Kočí and coauthors wrote a comprehensive review about photocatalytic reduction of carbon dioxide using TiO2 heterojunction. This review is well-written and organized. The CO2 reduction is currently a very hot topic with increased interest, and could possibly solve many fuels and environmental problems humans face. However, there are still some problems that must be addressed in this review, including missing references, lack of discussion, and typos. Moreover, the way authors referring and make figures and tables are very confusing and must be revised. Therefore, I suggest a major revision of this paper before publishing on Materials.
1. For all the Tables,  type of heterojunction column, the authors directly used the figures from the previous report. Although this way appears to be straightforward, the figure is small and hard to see in detail. Hence, I suggest authors use text to replace the figure, just saying type I, type II, or Z-scheme. Those figures in the tables must be regrouped and made into new figures.
2. The references of Table 1 is unclear. The references should be added for every entry.
3. Although the different pathways of photocatalytic reaction of CO2 are presented in Table 1, no detailed discussion was presented. The mechanism of CO2 reduction is important to make this review clear to the readers, and therefore I suggest authors add some discussion about the mechanism in Table 1. Besides, the influence of PH on reaction could also be discussed.
4. The heterojunction of TiO2/metal oxides and conjugated porous materials are an emerging class of materials that could possibly be used for CO2 photoreduction. The authors should discuss this in outlook and cite some related literature (Chem. Mater. 2021, 33, 15, 6158–6165; Angew. Chem. Int. Ed. 2020, 59, 6500– 6506).
5. Figure 1 is the same as Figure 2b. Besides, It is very inappropriate to just use a type-II scheme as an illustration of the electron-hole separation. The authors should draw a more general scheme to show the electron-hole separation for Figure 1.
6. Missing reference.
a. Reference is needed for “TiO2 has a relatively high recombination rate of photoinduced electron/hole (e− /h+ ) pairs (∼10 ns)” on page 2.
b. Reference is needed for “Consequently, the efficiency of e−/h+ separation in case of the p n heterojunction is quicker than that of type-II heterojunction photocatalysts because of to the synergic effect of the band alignment and the internal electric field.”
c. Reference is needed for “Unfortunately, this type of heterojunction photocatalysts have the one limitation, they can solely be used in the liquid phase.”
Here are some examples and authors should double-check the whole manuscript again.
7. Figure 3, please note which one is P-type and which one is n type semiconductor. Some introduction of n and p type semiconductors is also needed.
8. Direct Z-scheme without a mediator, type II and p-n junction heterostructures have similar energy diagrams. Could authors compare and discuss the difference of the three types and how to differentiate them in real experiments according to the previous reports?
9. For the discussion of detailed examples of photocatalysts, the author should refer them to the detailed figure and Table entry. For example, “Shi et al. [25] reported a yTiO2-x/…”. It is unclear which entry the authors referring to, and the corresponding figure must be provided. The authors missed this point in the whole manuscript. 
10. Grammar mistakes and typos. Here are some examples and authors should double-check the whole manuscript again.
a. Page 2. “More recently, several photocatalysts have been investigated for CO2 photocatalytic reduction, such as TiO2, g C3N4, ZnIn2S4, Bi2WO6, graphene (GR), CdS, SrNb2O6, and ZnO, however, the TiO2 is the most prevalently used due to its chemical stability, resistance toward corrosion, and mainly low cost [12].” There are two predicates in one sentence.
b. Page 4,  “had been photogenerated”. “Have” might be better.

Reviewer 2 Report

Presented review manuscript entitled “Photocatalytic Reduction of Carbon Dioxide on TiO2 Heterojunction Photocatalysts – A review” is written with good English, however, in some parts, the sentences are constructed in a chaotic way and must be corrected, e.g.:

- “The result is a constant increase in CO2 emissions and those of other greenhouse gases.”

- “Human emissions originate from the fossil fuels combustion, mainly coal, natural gas, and oil, along with soil erosion and deforestation.”

- “[…] action of sunlight […]”

- “In this case, under photocatalysis, […]”

- “[…] no products other (CH4, CH3OH, and HCOOH) than CO were detected, only a small amount of hydrogen.”

- “Recently, has been reported […]” etc.

Below, please find my detailed comments and questions regarding the manuscript:

- p-n heterojunction paragraph: “This diffusion of electrons and holes continues until the Fermi levels of semiconductors are not equal”. This statement is wrong. The diffusion continues until the Fermi levels ARE equal.

- Figure 4 contains “PS I” and “PS II” tags, while in text “PS-1” and “PS-2” appears. Please unify.

- Z-scheme paragraph: “Unfortunately, this type of heterojunction photocatalysts have the one limitation, they can solely be used in the liquid phase”. I strongly recommend adding a short explanation why, with suitable reference.

- Later in the Z-scheme description: “Figure 4c, shows the construction of this direct Z-scheme is the same as all-solid-state Z-scheme. In this case, the inactive charge carriers can be used by reacting with the acceptor/donor pair of acceptor/donor”. However, these sentences are contradictory to previous one (“[…] there is a combination of two different photocatalysts, without an electron mediator.”) and even Figure 4c. Please explain.

- Chapter 2.1., second paragraph. Authors first describe g-C3N4 as MCM, but later in the text use MCN. Please check and unify.

- Product name of Degussa commercial TiO2 is ones “P-25”, “p-25” or “P25”. Please check and unify.

- Page 8, second paragraph: “[…] TiO2 doped with amine species (N), and doped with metals, for example Nb, Ag, Au.”. If writing about Ag and Au, this is not doping, but surface modification. Authors themselves describe it correctly as surface modifications further in the text.  

- Some of the type of heterojunctions schemes given in Tables 2-7 are cut or simply illegible. The whole Tables also look chaotic. I suggest rotating of Tables horizontally, which will make them more readable.  

Concluding, I suggest a major revision of presented manuscript.

Reviewer 3 Report

The manuscript of Barrocas et al reviews the photocatalytic reduction of CO2 on TiO2 heterojunctions photocatalysts. The works is comphrehensive for this field and reviews a lot of aspects, including enough of the recent works in this field. However, some improvements can  still be made:

  1. Looking at the figures, Figure 1,2,3, 4 from the Introduction are all adapted from the same reference, ref 18.  Are there no other comphrensive reviews that can be used to corroborate these schematics? Also, some minor improvement can be made to the figures to improve readability.
  2. While the data in section 2 is comphrensive enough and separated in the different heterojunctions, some more separation between each type of heterojunction would be needed to further increase the readabiltiy of the text and capture the focus of the readers. E.g. section 2.1 has 4 pages and a half and than a 3 page Table. In addition the schematic from the Type of heterojunction in the Table is difficult to read. Authors could include each schematic related to the type of heterojunction as a separate figure where the work is discussed in the text of the manuscript. Similar for Section 2.2-2.6.
  3. The work is comphrehensive and reviews current achievment in this field, and the tables show a very good overview in the experimental parameters of CO2 reduction. The main dissadvantage of the current review lies in the difficulty to follow thorugh the extensive disccusion in sections 2.1-2.5. which can be improved by additional subheadings or separation in smaller parts, and in the difficulty to read and follow the heterojumvtion schematics from the 7 different Tables. For the latter, as previously mentiond, the schematics could be introduced as separate figures into the text.
  4. Other minor things: a) Figure 4 - abbreviation for PS I, and PS II should be included in the figure caption, the writing on the light (source) arrow is defficult to read (hν ≥ Eg).

Round 2

Reviewer 1 Report

The authors revised their manuscript with care and the current version has been highly improved. I would suggest publishing this paper. Here are some minor changes that the authors could make.

1. Figure 2a, should be "oxidation"

2. Section 2.7, the first sentence is a bit long. I suggest the authors could introduce a little bit about COF/metal oxides composites and then talk about the detailed example, like the authors did in other sections. Ref 94 and 95 could be separated. Ref 94 could be used for the first sentence about the introduction of the COF/metal oxides, and Ref 95 could be used for the specific example of Zhang. et.al.

3. Section 2.3, " Vis light absorption". Do you mean visible light absorption?

4. Table 4, entry 3, "1.0CuO-TiO2". The figure was not adopted well. Some part of CH3OH is missing, and there is an extra grey line on the left.

Reviewer 2 Report

Authors corrected the manuscript which is now more readable and correct. In this regard I suggest acceptance of the presented revised version. 

Reviewer 3 Report

The authors addressed the most critical comments. There are some more minor things to improve, for example:

- improve quality of some Z-shemes as they are still difficult to read (to increase text size or text is blurry). See for example:

  1. Table 2 - Z scheme for (0.3/1)TiO2/g-C3N4 --> text is blurry and not focused in the scheme
  2. Table 3 - Z scheme for CeO2/TiO2 (R-TiCe0.1)
  3. Table 5 - Z-scheme for CdS-TiO2-8
  4. Table 7 - Z scheme for 1:10 GaP/TiO2
